# Evidence of human influence on Northern Hemisphere snow loss

Alexander R. Gottlieb[1,2 ✉] & Justin S. Mankin[2,3,4]

Documenting the rate, magnitude and causes of snow loss is essential to benchmark the pace of climate change and to manage the differential water security risks of snowpack declines[1–4]. So far, however, observational uncertainties in snow mass[5,6] have made the detection and attribution of human-forced snow losses elusive, undermining societal preparedness. Here we show that human-caused warming has caused declines in Northern Hemisphere-scale March snowpack over the 1981–2020 period. Using an ensemble of snowpack reconstructions, we identify robust snow trends in 82 out of 169 major Northern Hemisphere river basins, 31 of which we can confidently attribute to human influence. Most crucially, we show a generalizable and highly nonlinear temperature sensitivity of snowpack, in which snow becomes marginally more sensitive to one degree Celsius of warming as climatological winter temperatures exceed minus eight degrees Celsius. Such nonlinearity explains the lack of widespread snow loss so far and augurs much sharper declines and water security risks in the most populous basins. Together, our results emphasize that human-forced snow losses and their water consequences are attributable—even absent their clear detection in individual snow products—and will accelerate and homogenize with near-term warming, posing risks to water resources in the absence of substantial climate mitigation.

Seasonal snow is regarded as a sentinel system for climate change. Warm winter temperatures can favour rain over snow and enhance snowmelt, reducing snow water storage and posing hydrologic risks to people and ecosystems[1–4]. Yet, puzzlingly, snow is not behaving as a sentinel (Fig. 1): although observations show consistent warming trends at the hemispheric, continental and river-basin scales (Fig. 1), there is no consistent pattern of snowpack loss across observational data products (Fig. 1b–e). As such, although the latest Intergovernmental Panel on Climate Change (IPCC) assessment concluded with high confidence that Northern Hemisphere springtime snow water equivalent (SWE; a typical measure of snow mass) has "generally declined" since 1981[7], it remains unclear where, when and by how much anthropogenic climate change has actually altered snowpack so far, especially at decision-relevant scales. Absent a robust attribution of human-forced snowpack changes, it is difficult to identify the regions most vulnerable to snow loss and, by extension, to develop appropriate strategies to manage present and future water security risks from snow changes.

At least three factors account for the inconsistent response of snowpacks to observed warming. Chief among them are the aforementioned observational uncertainties in estimates of SWE[5,6]. For example, in only one-third of the Northern Hemisphere's major river basins—and fewer than half of the dozen most populated—is there agreement across products on the direction of long-term snow change (Fig. 1c). Second, snowpack is highly variable across a range of timescales, reflecting low-frequency modes of climate variability, such as the Pacific Decadal Oscillation[8–10] or Atlantic Multidecadal Variability[11]. Disentangling the snowpack response to forcing thus also requires a robust estimate of regional snow responses to internal variability, such as those that come from initial condition large ensembles of climate simulations[12]. Attribution studies that rely on a small number of climate models and/or few model realizations (for example, refs. 13–16) may conflate internal variability and model structural uncertainties[17–19], the latter of which are quite large for snowpack[20–22], making attribution difficult. Lastly, the relationship between forcing and snowpack is not unidirectional: warming, for example, can enhance cold-season precipitation[23] and snowfall extremes[24], potentially offsetting warming-driven losses, particularly in cold, high-latitude or high-elevation regions[17,25]. Regional attribution studies (for example, refs. 13,14) have normalized SWE by cumulative cold-season precipitation in a rightful effort to reduce noise from precipitation variability and allow for a clearer identification of a temperature signal, but this strategy fails to capture the full effect of climate change on snow. Any attribution of human-caused snowpack declines must address these complications to be trustworthy and informative.

We address these uncertainties by combining an observations-based ensemble of snowpack, temperature, precipitation and runoff data products with empirical and climate models to attribute snowpack changes to anthropogenic warming at the hemispheric and river-basin scales. We use these insights to assess how changes to temperature and precipitation have affected snow water storage and to generalize how

[1]Graduate Program in Ecology, Evolution, Environment and Society, Dartmouth College, Hanover, NH, USA. [2]Department of Geography, Dartmouth College, Hanover, NH, USA. [3]Department of Earth Sciences, Dartmouth College, Hanover, NH, USA. [4]Division of Ocean and Climate Physics, Lamont-Doherty Earth Observatory, Columbia University, New York, NY, USA. ✉e-mail: alexander.r.gottlieb.gr@dartmouth.edu

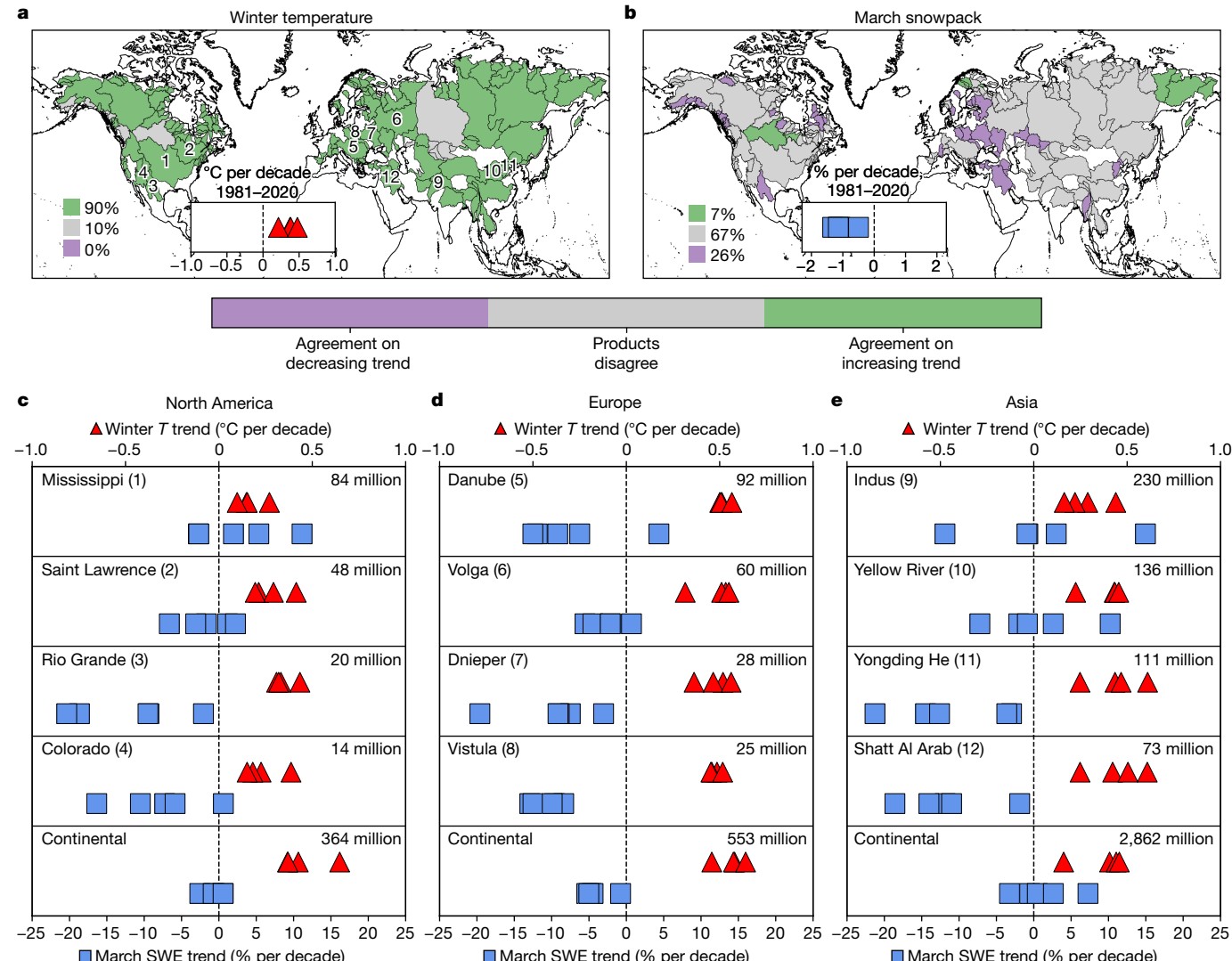

**Fig. 1 | Observed long-term warming trends are robust throughout the Northern Hemisphere, but snowpack trends are not. a**,**b**, Agreement across observational products (Supplementary Table 1) on the sign of trends in November–March average temperature (winter *T*, **a**) and March SWE (**b**) from 1981 to 2020. Numbers in bottom left show the percentage of basins with each category of agreement indicated on the colour bar. Insets: the hemispheric trends for each individual product. **c**–**e**, The trends for the four most populous river basins in North America (**c**), Europe (**d**) and Asia (**e**) that are generally considered snow dominated, as well as each continent (Methods). The locations of the basins are indicated on the map in **a**, corresponding to the number in parentheses. Temperature (red triangles) is referenced to the top *x* axis and SWE (blue squares) is referenced to the bottom *x* axis. The 2020 basin population is indicated in the top-right corner. Maps were generated using cartopy v0.18.0. River basin boundaries come from the Global Runoff Data Centre's Major River Basins of the World database[44].

snowpack and the runoff it generates will respond to additional warming. Together, our results provide a thorough documentation of the historical and future effects of climate change on snow water storage.

## A forced signal in snowpack observations

Despite the substantial uncertainty in spatially distributed estimates of snowpack (Fig. 1 and Extended Data Fig. 1), gridded snow products nevertheless share a distinct spatial pattern of historical trends that agrees well with in situ observations (Fig. 2a,b). Over the past 40 years, March SWE has sharply declined in the southwestern USA and much of western, central and northern Europe by 10% to 20% per decade. Strong snow decreases extend eastwards across the Eurasian continent into parts of central Asia, per the gridded products (Fig. 2b and Extended Data Fig. 1), although a lack of in situ reference points there makes it difficult to validate these trends. In contrast, the cold continental interiors of central North America and northern Eurasia have seen increasing

spring snowpacks, with in situ observations indicating a deepening of over 20% per decade in the Northern Great Plains and parts of Siberia, whereas gridded products indicate more modest increases of 5% to 10% per decade. Snow-dominated regions that lack in situ observations, such High Mountain Asia and the Tibetan Plateau, show weak trends in the gridded observational ensemble mean (Fig. 2b), which belie directionally inconsistent trends in individual data products (Fig. 1b,e and Extended Data Fig. 1).

Coupled climate model simulations forced with historical human and natural forcing capture some features of the observed historical spatial pattern of snow change, particularly the large snow loss over most of Europe and modest gains over Northern Eurasia (Fig. 2c and Extended Data Fig. 2). The historical climate model experiments capture parts of the spatial structure of snow change over North America, including declines in the southwest and northeast, but show modest declines in the continental interior where observations report deepening snowpacks (Extended Data Fig. 2). Meanwhile, simulations that

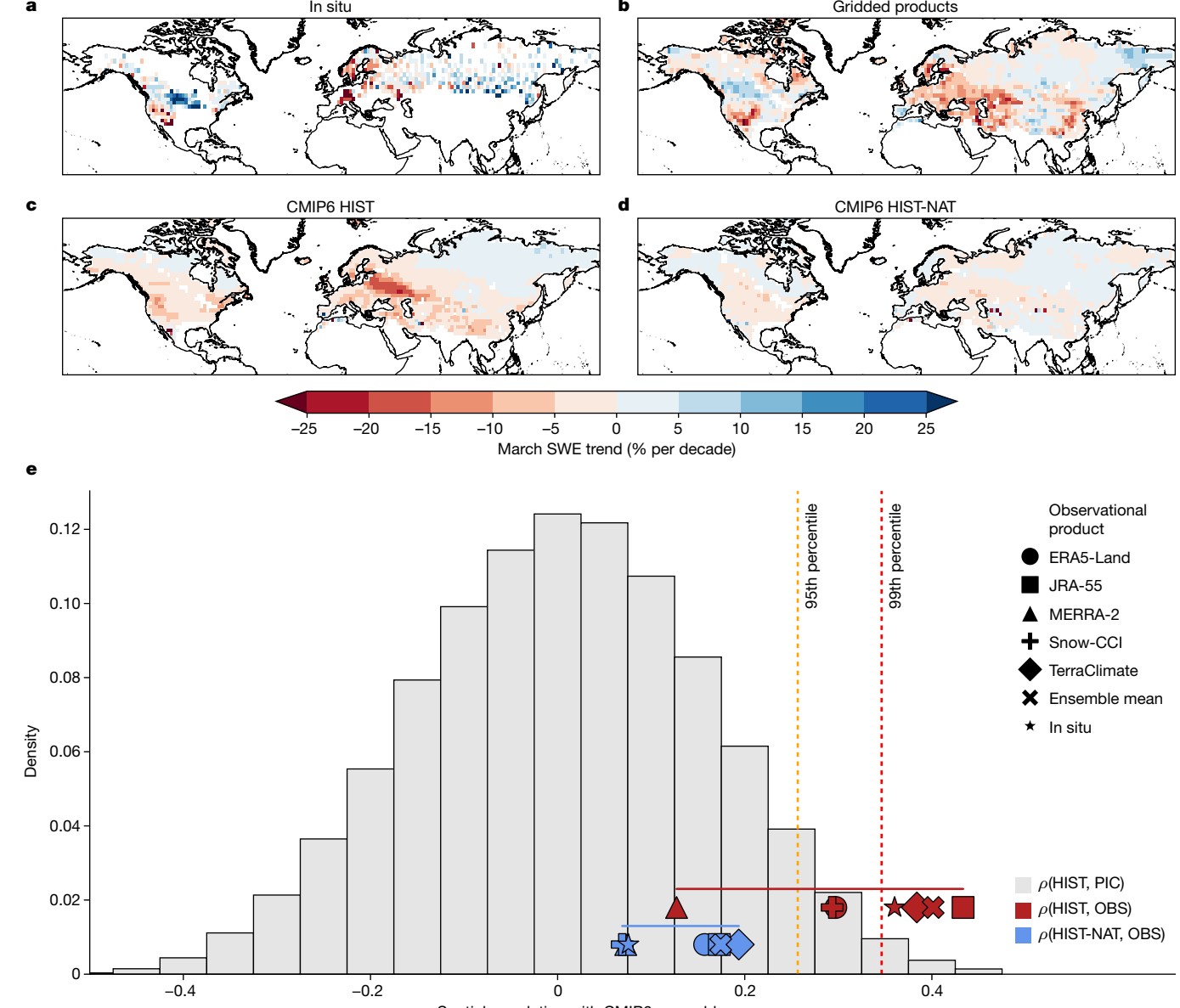

**Fig. 2 | Climate model experiments reveal that human-caused warming has influenced Northern Hemisphere snowpack trends. a**–**d**, Trend in March SWE from 1981 to 2020 in in situ observations (**a**), the ensemble mean of five long-term gridded SWE products (**b**), and the multimodel mean of CMIP6 historical simulations with (**c**) and without (**d**) anthropogenic emissions. **e**, Spatial pattern correlation ($\rho$) of 1981–2020 March SWE trends between the CMIP6 multimodel mean HIST (red symbols) and HIST-NAT (blue symbols) simulations and each observational (OBS) SWE product (see legend). The grey histogram indicates the empirical probability density function of spatial correlations between trends from the historical simulations and all possible 40-year trends from unforced pre-industrial control (PIC) simulations ($N = 78,601$). The red (orange) vertical dashed line indicates the 99th (95th) percentile of this empirical distribution. Maps were generated using cartopy v.0.18.0.

exclude anthropogenic emissions fail to capture the observed pattern of snow change (Fig. 2d).

To be able to claim that human interference in the climate system is responsible for the observed hemispheric pattern of snowpack trends, we calculate the chances that the observed pattern of snow change could have arisen from natural climate variability alone. We follow a widely used attribution approach[26–29] and generate a distribution of pattern correlations between 40-year SWE trends from forced (historical or HIST) and unforced (pre-industrial control or PIC) climate model simulations (Methods). This exercise provides a null distribution (the grey background histogram in Fig. 2e) indicating how much a spatial pattern of SWE trends arising from model-simulated natural variability alone could resemble a pattern consistent with those that include anthropogenic forcing. We then correlate the spatial pattern of SWE

trends in each observational dataset with those from the ensemble mean of two different climate experiments: the HIST simulations (red symbols in Fig. 2e), representing historical anthropogenic forcing and the historical-nat, or HIST-NAT, simulations (blue symbols in Fig. 2e), representing a historical climate without human-caused greenhouse gas emissions. Finally, we compare the observed correlations to the null distribution to calculate the probability that the degree of similarity between the observations and HIST and HIST-NAT simulations could have arisen from natural variability.

We find that, in the language of the IPCC, it is virtually certain (>99% probability) that human emissions have contributed to the observed pattern of March snowpack trends in in situ observations and in the average of the gridded ensemble, as well as in the TerraClimate reanalysis and the Japanese 55-year Reanalysis (JRA-55). We note that the strength

of this claim is subject to the choice of dataset, as the ERA5-Land reanalysis (97%) and the satellite remote sensing-based Snow-CCI product (97%) show a slightly lower, but still an 'extremely likely' probability, and there is no detectable influence when examining the MERRA-2 reanalysis (78%). Thus, despite the substantial observational uncertainty in long-term snow trends among data products, there seems to be a shared structure in the spatial pattern of observed change that is consistent with that from anthropogenic forcing. Crucially, this similarity is absent when these products are compared with simulations that include only solar and volcanic forcing on the climate system (HIST-NAT; blue symbols in Fig. 2e), as not a single pattern is distinguishable from natural variability. As such, we can considerably strengthen the recent IPCC claim about snow trends and say with a high degree of confidence that human emissions have contributed to the observed pattern of spring snowpack trends across the Northern Hemisphere over the past 40 years.

## River-basin-scale snowpack changes

The coupled climate model experiments such as those presented in Fig. 2 are a powerful tool for detecting and attributing human influence on the broad features of the hemispheric pattern of SWE trends. Yet the ability of these models to capture the magnitude and detailed spatial structure of observed trends is limited (see the range of the $x$ axis in Fig. 2e), undermining the ability to assess forced snow change and its consequences at impact-relevant scales. To that end, we pursue a data–model fusion approach using a random forest machine-learning algorithm that has been applied in a wide variety of attribution contexts[8,30–34], where we combine empirical models of SWE with climate model simulations to allow us to flexibly estimate how anthropogenic emissions have affected the temperature and precipitation that drive SWE at finer scales (Methods). We combine a number of gridded snowpack, temperature and precipitation datasets (Extended Data Table 1) in an effort to produce an ensemble of empirical reconstructions of historical March SWE at the basin scale (Methods) that skillfully reproduce observed trends and variability in those datasets, with the spatial pattern correlations of reconstructed and observed trends ranging from 0.9 to 0.97 (Extended Data Fig. 3) and a median root-mean-square error (RMSE) across all products and basins of under 8% (Extended Data Fig. 4). Furthermore, the snowpack reconstruction models are able to skillfully hindcast long-term trends and variability in out-of-sample in situ snow data, with a trend pattern correlation across roughly 3,000 sites of 0.72 and a median RMSE of 22% (Extended Data Fig. 5).

Our strategy to empirically reconstruct basin-scale SWE many times using a large number of dataset combinations has three goals. First, we want to be able to effectively sample the observational uncertainty in snow and climate that has undermined snow attributions so far (Fig. 1). Second, we need to reconstruct snowpack as a function of temperature and precipitation to isolate how forced and unforced changes in those quantities have shaped observed snowpack changes at impact-relevant scales. Our ensemble of empirical snowpack reconstructions give us the experimental control to assess the drivers of snow changes. Lastly, we want to be able to assess whether signals of forced snowpack changes emerge above the noise of observational, internal variability and climate model uncertainties, and to quantify those sources of uncertainties to improve snowpack constraints[18] (Extended Data Fig. 7). By using all factorial combinations of observations and climate models, we can fully characterize and quantify these sources of uncertainty and achieve a better estimate of the true forced signal than could be achieved with any single dataset[5,32].

Our ensemble of observations-based reconstructions of March SWE (Fig. 3 and Extended Data Fig. 4) shows that spring snowpack has declined over the past four decades in many mid-latitude basins, with modest increases in cold, high-latitude basins (Fig. 3a). The largest decreases of around 10% per decade are seen in the river basins of the southwestern USA and Europe, in agreement with the long-term trends from in situ SWE measurements there[35,36]. Despite the substantial uncertainty in March SWE trends in the gridded observational products themselves (Fig. 1), our empirical reconstructions show a consistent direction of trends in about half of all major river basins (82 out of 169). At the same time, however, there are large concentrations of basins with insignificant March SWE trends in High Mountain Asia, northern North America and Siberia (outside of the Far East, where increases similarly agree with in situ observations[37]) driven largely by disagreement on the direction of trends across the ensemble of SWE reconstructions.

The value of our basin-scale SWE reconstructions is that they allow us to isolate the influence of anthropogenically forced trends in temperature and precipitation on snowpack trends at hydrologically relevant scales while fully sampling observational, empirical and climate model uncertainties. We difference the Coupled Model Intercomparison Project Phase 6 (CMIP6) HIST and HIST-NAT experiments to estimate the forced response of temperature and precipitation. We then remove that from the observed temperature and precipitation time series and re-estimate our snowpack reconstructions, giving us an ensemble of counterfactual no-anthropogenic-climate-change snowpack (Methods). Although fewer than a quarter of all basins (37 out of 169) show significant counterfactual trends (Supplementary Fig. 1), some basins, such as the Rio Grande (6.3% per decade), still show consistent declines over the past 40 years, even without human interference with the climate. Such declines are consistent with regional teleconnections to low-frequency oceanic variability, such as the Pacific Decadal Oscillation[10], which can drive decadal-scale hydroclimate trends in these regions independent of those from anthropogenic warming.

We note that the CMIP6 models tend to over-estimate the historical warming trend compared with observations in some regions, particularly over central North America and eastern Europe (Extended Data Fig. 6 and Supplementary Fig. 2). At the same time, however, fewer than 1% of apparent biases over the hemisphere fall outside the range of model internal variability, suggesting that models are skillfully capturing Northern Hemisphere winter land-temperature trends[38]. The models also underestimate the multidecadal drying in the southwestern USA, which has seen historical precipitation declines driven by both internal ocean–atmosphere variability and anthropogenic forcing[8], and underestimate observed wetting over the Tibetan Plateau (Extended Data Fig. 6 and Supplementary Fig. 2). Once again, however, fewer than 3% of precipitation biases lie outside that possible from modelled internal variability, suggesting these biases do not undermine our attribution.

Our approach sifts through the observational and model noise to reveal that human-forced changes to temperature and precipitation have altered spring snowpack trends in 31 major river basins across the Northern Hemisphere (Fig. 3e). The spatial pattern of forced SWE trends is similar to the historical trends (compare Fig. 3a and 3e), with anthropogenic climate change having reduced spring snowpacks in the mid-latitudes (basins south of 60° N) by 4.1 ± 3.4% per decade (mean ± s.d.) and enhanced them in the cold, high-latitude basins that drain into the Arctic Ocean by 2.5 ± 1.8% per decade (Fig. 3e). Interestingly, we are able to detect a forced SWE decline in major basins such as the Columbia (4.8% per decade) where historical observations indicate modest increases since 1981 or the Saint Lawrence (6.9% per decade), where observed trends have been small and statistically insignificant. These examples suggest that internal variability in the climate system has been masking large forced snowpack reductions in some regions[17]. Likewise, there are basins like the Rio Grande, which have suffered large historical snowpack declines of over 10% per decade, but for which there is little agreement that forced temperature and precipitation changes have caused those declines, reinforcing the notion that low-frequency variability can overwhelm forced signals in snow and hydroclimate, even on multidecadal timescales[17,39]. Indeed, internal variability is the dominant source of uncertainty in the magnitude

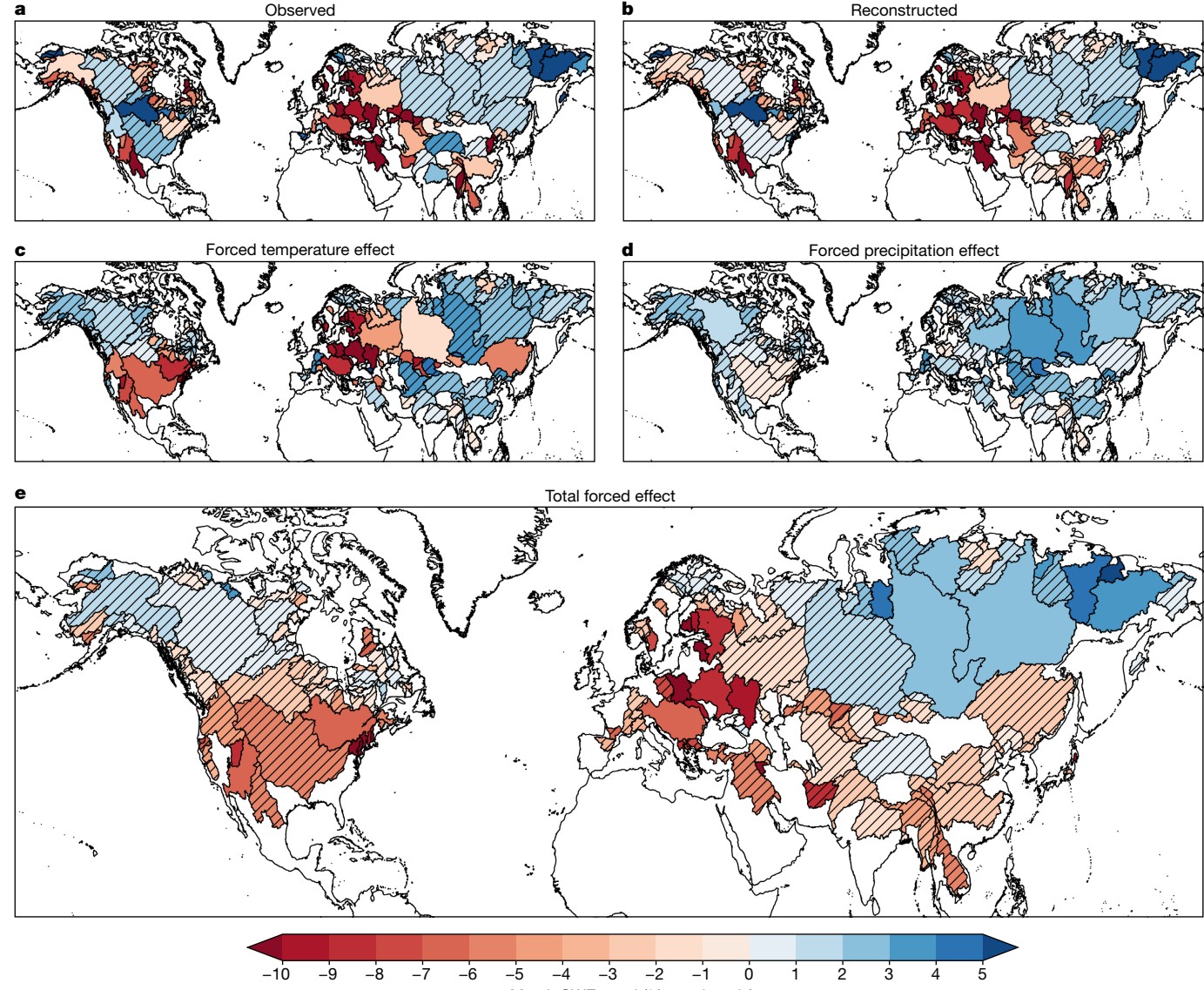

**Fig. 3 | Empirical snowpack reconstructions reveal the countervailing effects of human-forced temperature and precipitation trends on basin-scale snow changes. a**, Average observed 1981–2020 March SWE trends from 5 long-term SWE data products in 169 major Northern Hemisphere river basins. **b**, As in **a** but for our observation-based reconstructions. **c**, Effect of anthropogenically forced temperature changes on March SWE trends, given by the ensemble mean difference between the statistically reconstructed historical trend and the reconstructed trend with forced changes to temperature removed. **d**, As in **c** but for forced precipitation changes. **e**, As in **c** and **d** but for forced changes to both temperature and precipitation. The hatching indicates basins where fewer than 80% of observations or reconstructed estimates agree on the sign of the trend or forced effect. Maps were generated using cartopy v.0.18.0. River basin boundaries come from the Global Runoff Data Centre's Major River Basins of the World database[44].

of forced response—over climate model structural differences and observational uncertainty in SWE, temperature and precipitation—in roughly one in eight basins (Extended Data Fig. 7).

Our isolation of the effects of forced changes in temperature (Fig. 3c) and precipitation (Fig. 3d) show that anthropogenic temperature changes have generally reduced March SWE across the hemisphere, except in the coldest basins, although uncertainty in the underlying SWE observations and in the regional temperature response of the climate models limits agreement over much of northern North America and Asia (Fig. 2c and Extended Data Fig. 7). Anthropogenically forced precipitation increases have offset some warming-driven losses (Fig. 3d) consistent with observed human-caused increases in winter precipitation in many of the Northern Hemisphere's cold regions[23]. Outside of cold continental interiors[40], however, forced snowpack increases from precipitation are generally insignificant, reflecting

both the greater model uncertainty in precipitation and the larger contribution of internal variability to hydroclimate uncertainty[19,41].

## Nonlinear sensitivity of snow to warming

Disentangling forced from unforced snow changes (as presented in Fig. 3) is essential to inform decisions to manage present and future snow loss. Our analysis makes clear that there is indeed a fingerprint of anthropogenically forced SWE trends across the Northern Hemisphere and that for some regions, natural variability has been sufficient to mask or reverse snow trends. But such an analysis is not just valuable for what it says about snow changes so far. It is valuable because it helps reveal the highly nonlinear sensitivity of snowpack to warming (Fig. 4), and in doing so, resolve the conundrum of why it is that—despite warming—there has not been a commensurate decline in snow water

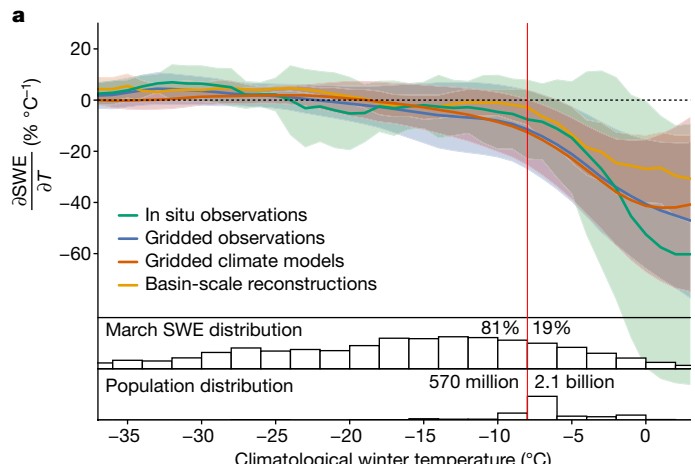

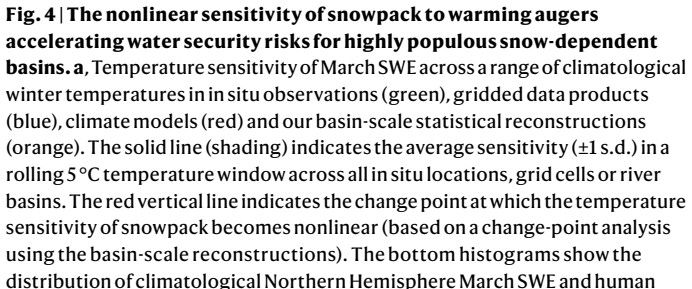

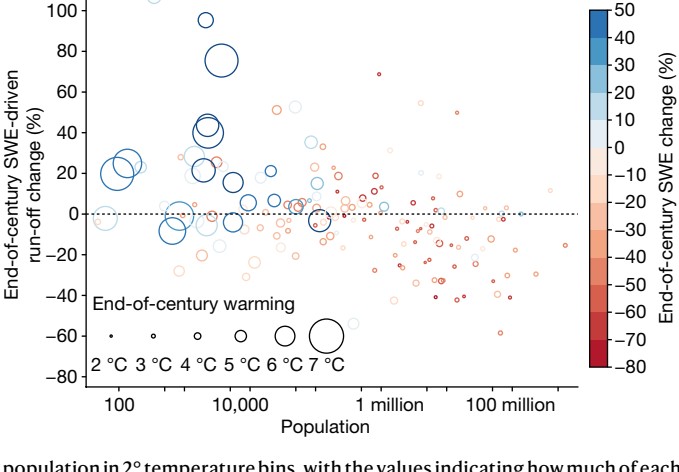

**Fig. 4 | The nonlinear sensitivity of snowpack to warming augers accelerating water security risks for highly populous snow-dependent basins. a**, Temperature sensitivity of March SWE across a range of climatological winter temperatures in in situ observations (green), gridded data products (blue), climate models (red) and our basin-scale statistical reconstructions (orange). The solid line (shading) indicates the average sensitivity (±1 s.d.) in a rolling 5 °C temperature window across all in situ locations, grid cells or river basins. The red vertical line indicates the change point at which the temperature sensitivity of snowpack becomes nonlinear (based on a change-point analysis using the basin-scale reconstructions). The bottom histograms show the distribution of climatological Northern Hemisphere March SWE and human

population in 2° temperature bins, with the values indicating how much of each distribution falls on each side of the change point. Temperatures on the *x* axis are the average November–March temperature over the 1981–2020 period from each in situ location or grid cell. Only climatologically snow-covered grid cells are used to calculate the basin-average temperature. **b**, Percentage change in basin-scale March SWE-driven April–June runoff in 2070–2099 under SSP2-4.5 relative to 1981–2020 (Methods) versus basin population. The dots are coloured by the percentage change in March SWE in 2070–2099 relative to 1981–2020 and sized by the CMIP6 ensemble mean projected end-of-century temperature change.

storage across the Northern Hemisphere (for example, Fig. 1). It also makes clear why we should expect snow losses to rapidly accelerate, with widespread water security consequences (Fig. 4b).

Examining the shape of the relationship between average winter temperatures and the marginal sensitivity of snow change to temperature change clarifies why snow detection has been elusive so far and why even modest levels of warming suggest much sharper snow declines to come (Fig. 4a). The responsiveness of snow to 1 °C of warming depends on climatological winter temperatures. Below historical temperatures of about −8 °C (determined from change-point analysis), spring snowpack is little affected by warming; however, each additional 1 °C of warming beyond that point results in accelerating losses.

There are several notable features in these curves. First, is their scale and data invariance: the location of the inflection point in temperature sensitivity is consistent when it is estimated from point measurements, gridded data products, climate models or our basin-scale reconstructions. This consistency suggests that despite substantial measurement and modelling uncertainties, simple thermodynamics can explain much of snow's historical and future response to warming. As the climatological temperature of a location warms towards the freezing point, the likelihood of subseasonal temperatures exceeding thresholds where precipitation is partitioned towards rain over snow or accumulated snowpack will melt increases exponentially. We note, however, that these thresholds themselves are not constant in space, owing to factors such as topography and distance from oceanic moisture sources[42], which may account for some of the uncertainty in snow sensitivities at any one climatological temperature (shading in Fig. 4a). Second, the marginal sensitivity of snow to temperature change provides some intuition for the spatial pattern of SWE trends shared by the observations and climate models in Fig. 2: in general, the largest snowpack declines are seen in the climatologically warmest places, which sit just beyond the inflection point in the curve presented in Fig. 4a. There, small increases in temperature have led to large declines in snowpack. In contrast, cold regions see little change or in some cases, increased SWE. Such locations sit on the flat, insensitive part on the curve

defining the relationship between climatological temperatures and snow sensitivity (Fig. 4a).

Lastly, the fact that snow is relatively insensitive to warming below climatological winter temperatures of about −8 °C helps explain the lack of clear snow trends at the hemispheric scale despite substantial warming so far: over 80% of the March snow mass in the Northern Hemisphere is found in places to the left of this inflection point (upper inset distribution, Fig. 4a). In those regions, warming has little effect. Notably, much of the 20% of hemispheric snow mass remaining resides just to the right of the −8 °C inflection point, hovering near a snow-loss cliff, where marginal increases in temperature imply larger and larger snow losses to come. What is clear is that in these regions, snow declines so far have been relatively small compared with natural variability. Indeed, the likelihood of observing a statistically significant trend in SWE begins increasing around this inflection point in climatological temperature (Supplementary Fig. 3). Such a relationship suggests that further warming and thus additional time spent beyond this −8 °C threshold will homogenize snow trends towards more consistent declines, portending widespread and accelerating snow losses for many basins over the coming decades.

Crucially, the highly nonlinear relationship between snow sensitivity and climatological temperature implies rapidly emerging water security risks to people. Although 80% of the Northern Hemisphere's snow mass is found in cold places that have historically been insensitive to warming, 80% of the hemisphere's inhabitants reside in the snow-dependent regions beyond this inflection point (lower inset distribution, Fig. 4a). As such, further warming is likely to have rapidly emerging impacts on snow water resources in the mid-latitude basins where people reside and place competing demands on fresh water.

To assess this, we consider the population exposure to both projected snow loss and attendant spring snowmelt driven runoff change (Fig. 4b). Under Shared Socioeconomic Pathway (SSP) 2–4.5, a 'middle-of-the-road' emissions scenario, the most highly populated basins are expected to see strong declines in spring runoff as a result of nonlinear snow loss, even in the face of relatively modest warming projected in those regions (Fig. 4b and Extended Data Fig. 8).

The western USA, for example, is poised to see particularly sharp spring runoff declines in the upper Mississippi (84 million people, 30.2% spring runoff decline), Colorado (14 million, 42.2%), Columbia (8.8 million, 32.7%) and San Joaquin (6.8 million, 40.9%) river basins (Extended Data Fig. 8). The most populous basins in Europe, such as the Danube (92 million, 41.0%), Volga (60 million, 39.5%), Rhine (51 million, 33.0%) and Po (18 million, 40.5%) could face water-availability challenges of a similar magnitude. Future changes to SWE-driven spring runoff in Asia, the continent with the greatest number of people living in snow-influenced basins, show substantially less agreement (hatching in Extended Data Fig. 8). Snowpack in cold and sparsely populated basins, meanwhile, is likely to be resilient to high levels of winter warming exceeding 5 °C, such as that arising from Arctic amplification[43], and the coldest may see increased snowpacks and enhanced spring runoff into the Arctic Ocean of over 10% on average (Fig. 4b and Extended Data Fig. 8).

## Managing and leveraging snow uncertainty

Our analysis uses snowpack observations, climate models and an observations-based ensemble of snowpack reconstructions to attribute changes in spring snow water storage at the hemispheric and river-basin scales. Our results explain why snowpack has been a poor sentinel system to assess the pace and magnitude of global warming so far, but why despite that, we should expect unprecedented snowpack declines with only modest additional warming. There is a highly nonlinear temperature sensitivity of snowpack, foreshadowing marked reductions in spring snowpack and associated snow-driven runoff in highly populated basins where snowmelt has an important role in water supply. Our analysis reveals that many of the world's most populous basins are hovering on the precipice of rapid snow declines and that such losses may only be detected across all observational data products once the water security impacts of snow loss have already manifested. Thoughtful adaptive planning and risk mitigation—particularly around capital-intensive and contentious infrastructure to manage winter flood risks coupled with reduced warm-season streamflow—requires advance warning. The highly nonlinear marginal sensitivity to snow we identify clarifies why such warning in the observations so far has been elusive, and also why waiting until the impacts manifest could be too late to effectively manage their risks. Such warning, we show, will probably only come from the observations once warming is sufficient to push regions into this highly nonlinear snow-loss regime.

We emphasize that we can report these findings to provide meaningful warning because of—rather than despite—uncertainty. Snow datasets may not agree with one another on the magnitude of snowpack or its variability and long-term trends through time (Fig. 1a and Extended Data Fig. 1). Yet in situ measurements and all gridded data products, apart from one, show a spatial structure consistent with anthropogenic forcing of the climate system. The consistency across diverse datasets allows for a much higher degree of confidence in the identification of forced snowpack trends than could be achieved using a single snow dataset alone. Furthermore, the lack of precise knowledge about the true state of snowpack over time, cold-season temperature and precipitation, and their response to anthropogenic emissions allows us to leverage multiple sources of uncertainty to produce over 12,000 estimates of the effects of anthropogenic climate change on spring snowpack in each of the major river basins of the Northern Hemisphere and identify a statistically stable estimate of the forced signal.

In addition, there is value in identifying and quantifying these sources of uncertainty in forced snowpack changes (Extended Data Fig. 7), as it can guide future scientific and operational decision-making[18]. For instance, uncertainty in the forced response of temperature and precipitation arising from structural differences between climate models is the dominant source of uncertainty in the magnitude of forced March SWE trends in over half (95 out of 169) of all basins (Extended

Data Fig. 7), suggesting that improving the skill of climate models in capturing regional climate would go a long way towards constraining historical and future snow change. Uncertainty in SWE data products themselves is also a limiting factor in many basins where in situ observations are sparse or non-existent (Extended Data Fig. 7), suggesting that constraining observational estimates of SWE would be valuable. Finally, identifying the contribution of irreducible uncertainty in SWE trends from internal variability in the climate system (Extended Data Fig. 7) is also essential, as it indicates the range of physically consistent snowpack trajectories for which water resource managers and stakeholders must be prepared[17,18].

Together, our findings portend serious water-availability challenges in basins where snowmelt runoff constitutes a major component of the water supply portfolio. Improving our understanding of where and how climate change has and will affect snow water resources is vital to informing the difficult water resource management decisions that a less snowy future will require.

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

# Methods

We use two approaches to evaluate the effects of anthropogenic climate change on spring snowpack. First, we follow an attribution that uses the correlation between observed historical snowpack trends from several SWE data products and those from climate model simulations. Second, we take a data–model fusion approach in which we generate a large observation-based ensemble of historical snowpack and estimate what March SWE would have been in the absence of anthropogenically forced changes to cold-season temperature and precipitation. The former indicates forced changes to hemispheric snowpack and the latter indicates forced snow changes at hydrologically relevant scales.

## Data

Our ensemble of SWE observations consists of five long-term gridded datasets from the European Center for Medium-Range Weather Forecasting's (ECMWF) ERA5-Land reanalysis[45]; the Japan Meteorological Agency's JRA-55 reanalysis[46]; NASA's MERRA-2 reanalysis[47]; the European Space Agency's Snow-CCI, Version 2.0[48]; and TerraClimate[49]. Products with a submonthly temporal resolution are averaged across all available March values. We focus on March because it is climatologically the month of maximum snow mass in the Northern Hemisphere[20] and there is an extensive collection of in situ measurements taken during March against which we can benchmark our results. Because the satellite remote-sensing-based Snow-CCI product is masked over mountainous terrain, we follow the approach of ref. 20 and fill SWE values in mountainous cells with the mean value from the other four data sources. For non-mountainous grid cells, we use the unaltered Snow-CCI data. In addition, we use in situ SWE data from the Snowpack Telemetry Network (SNOTEL) network in the western USA[50]; the Canadian historical Snow Water Equivalent dataset (CanSWE)[51]; and the Northern Hemisphere Snow Water Equivalent (NH-SWE) dataset, a hemispheric dataset that converts far more abundant snow depth observations to SWE using a well validated model[52]. Only in situ observations with records for at least 35 years between 1981 and 2020 are retained, resulting in a set of 550 from SNOTEL, 341 from CanSWE and 2,119 from NH-SWE.

Gridded precipitation data come from the ECMWF's ERA5 reanalysis[53]; the Global Precipitation Climatology Centre (GPCC)[54]; MERRA-2[47]; Multi-Source Weighted-Ensemble Precipitation (MSWEP), Version 2[55]; and TerraClimate[49]. Gridded temperature data come from Berkeley Earth (BEST)[56]; NOAA's Climate Prediction Center (CPC) Global Unified Temperature[57]; ERA5[53]; and MERRA-2[47]. Daily gridded runoff data come from the ECMWF's Global Flood Awareness System (GloFAS)[58]. Details of all datasets used in the analysis are given in Extended Data Table 1.

For the climate-model-based attribution and observation-based reconstructions, we regrid all data to 2° × 2° and 0.5° × 0.5° horizontal resolution, respectively, using conservative regridding. For all data except runoff, grid cells where March SWE is zero in more than half of all product years are masked out, as is Greenland.

We also use climate model output from 12 models that archived monthly SWE ('snw') data from the pre-industrial control (PIC), historical (HIST), historical-nat (HIST-NAT) and SSP2-4.5 CMIP6 experiments, as well as monthly air temperature ('tas') and precipitation ('pr') data from the HIST, HIST-NAT and SSP2-4.5 experiments[27,28]. All model output are regridded and masked as with the gridded observational data. Consistent with the Detection and Attribution Model Intercomparison Project (DAMIP) protocol, the HIST simulations, which end in 2014, are extended to 2020 using the SSP2-4.5 scenario[59]. For simplicity, 'historical' (HIST) will always refer to these extended time series. Model details are given in Extended Data Table 2.

To provide estimates of hydrologic quantities at decision-meaningful scales, we aggregate from the gridded to the river-basin scale using basin extents from the Global Runoff Data Center's Major River Basins of the World database[44]. All empirically estimated grid-cell values of SWE, precipitation and runoff (in mm, or equivalently kg m$^{-2}$) are multiplied by the grid cell area (in m$^2$) before summing all grid cells within a basin to calculate basin-scale mass (in kg). Basin- and hemisphere-average temperatures are given by the area-weighted mean temperature of all snow-covered grid cells.

All estimates of basin population are calculated using the 2020 values from the 15 arcmin Gridded Population of the World, Version 4 (GPWv4) dataset from NASA's Socioeconomic Data and Applications Center[60].

## Attributing SWE trends to anthropogenic forcing

Our hemispheric attribution approach tests whether the similarity between observed and climate-model-simulated forced SWE trends exceeds what could be possible from natural climate variability alone[26–29]. To evaluate the null hypothesis that the pattern of SWE trends in the HIST simulations could be the result of natural variability alone, we calculate the spatial pattern of trends in March SWE from 1981 to 2020 in each model's HIST simulation and for every unique 40-year period from those same models' unforced PIC simulations (for example, for a 500-year PIC simulation, we generate 461 maps of 40-year trends). All trends are calculated using the Theil–Sen estimator, a non-parametric technique for estimating a linear trend that is more robust to data that is skewed or contains outliers than trends calculated using ordinary least squares regression. Then, we calculate the Spearman (rank) correlation coefficient between the spatial maps of HIST and PIC trends to quantify the pattern similarity. The resulting empirical distribution of 78,601 correlations (background histogram on Fig. 2) represents the likelihood that the pattern in the forced historical simulations could have arisen from natural variability alone.

We quantify the similarity between the observed pattern of SWE trends and the model-estimated response to forcing by taking the Spearman spatial correlation between the map of trends from each observational product and the multimodel mean map from the HIST simulations (red symbols in Fig. 2e). For this analysis, the in situ observations are aggregated to the same 2° × 2° grid as the gridded observations and climate models by taking the mean trend of all stations within each grid cell (Fig. 2a). If the correlations between the observations and HIST simulations are greater than almost all of the correlations between the HIST and PIC simulations, we can reject the null hypothesis that the observed historical pattern could have arisen from natural variability alone and claim that a response to historical forcing is present in the observed pattern. Furthermore, if we cannot reject the null hypothesis using the correlations between the observations and HIST-NAT simulations with only solar and volcanic forcing, then it is unlikely that the observed pattern is the result of natural radiative forcing. Combined, these two lines of evidence strongly indicate that anthropogenic forcing is causing the observed patterns of SWE trends.

## Observation-based snow reconstructions

As another means of attributing historical SWE change, and to better understand its patterns and drivers at scales more commensurate with the impacts of snow loss, we generate a large observation-based ensemble of historical March SWE with and without the effects of anthropogenic forcing. We do so by using the common random forest machine-learning algorithm, which fits randomized regression trees on bootstrapped samples of the data and averages their predictions together. The decision tree framework is particularly well suited to pick up nonlinear interactions, such as that between temperature and precipitation in the context of snow, as well as correlated predictors. The random forest algorithm has been applied to reconstruct a wide variety of biogeophysical variables that are shaped by temperature, precipitation and their interaction, including historical runoff[61], crop yields[62] and climate-induced species range shifts[63]. In each instance, the random forest model was found to significantly outperform both other machine-learning algorithms and more traditional approaches such as linear regression. In addition, for this particular application of reconstructing historical snowpack, the model imposes no prior

assumptions about temperature thresholds for rain–snow partitioning or snowmelt, which can vary substantially in space and are themselves a contributor to uncertainty in modelled estimates of SWE[42,64]. We model March SWE as a function of average monthly temperature and cumulative monthly precipitation from the previous November to March:

$$SWE_{y,i} = f(T_{y,11,i}, P_{y,11,i}, T_{y,12,i}, P_{y,12,i}, T_{y,1,i}, P_{y,1,i}, T_{y,2,i}, P_{y,2,i}, T_{y,3,i}, P_{y,3,i}) \quad (1)$$

where $SWE_{y,i}$ is average March SWE in water year (October–September) $y$ at grid cell $i$, $f$ is the random forest model, $T_{y,m,i}$ is the average temperature in month $m$ of water year $y$ and grid cell $i$, and $P_{y,m,i}$ is the total precipitation in month $m$ of water year $y$ and grid cell $i$. We fit the model using the full spatiotemporal panel of $0.5° \times 0.5°$ gridded data (that is, all grid-cell years from 1981 to 2020), then aggregate the predicted gridded values to the river-basin scale. We find that training a single model on the full panel of data offers two main advantages over training multiple models on more local data (for example, a model for each river basin). First is that the out-of-sample prediction skill of the full panel model is significantly higher in many highly populated mid-latitude basins of the western USA, western Europe and High Mountain Asia; local models are more skilful in fewer than 20% of basins, concentrated in sparsely populated high-latitude basins where the skill of the full panel model is already high (Extended Data Fig. 3). Second, training a single model on data from the entire hemisphere provides greater statistical stability of projections made with large perturbations to the input variables, such as adding an end-of-century climate change signal (Extended Data Fig. 8), which could exceed the support of local historical observations as records fall at an increasing rate[65,66].

To adequately sample and quantify the observational uncertainty in snowpack, temperature and precipitation and create a sufficiently wide ensemble of possible SWE values, we repeat this procedure for all combinations of 6 SWE (5 gridded + in situ), 4 temperature and 5 precipitation datasets (Extended Data Table 1), providing 120 ($6 \times 4 \times 5$) estimates of basin-scale March SWE from 1981 to 2020. Our ensemble approach is motivated by two main considerations. First, it is difficult to determine what represents 'true' snowpack at hydrologically relevant scales. All methods of estimating spatially distributed snowpack (for example, remote sensing or reanalysis) have their intrinsic limitations that result in high levels of disagreement on snow mass, its variability and long-term trends[5,6], as we show in Fig. 1. In situ measurements may represent truth at the locations at which they are collected, but are difficult to generalize, especially in complex terrain. As a result, using these point observations to adjudicate which gridded products (whose values represent averages over tens to tens of thousands of kilometres) lie closest to 'truth' is challenging. Given the inability to know the true state of snowpack or rigorously rule out any of its various gridded estimates, we choose to consider these observational products as equally valid estimates of truth in which we can attempt to identify shared responses. Second, the ensemble approach allows us to capture the structural uncertainty in how SWE responds to changes in temperature and precipitation, which are themselves subject to data uncertainties (Supplementary Fig. 2). Using all dataset combinations, we can sample and characterize uncertainty in SWE, temperature and precipitation and their covariance with one another. Such an approach has been used to estimate forced changes in components of the Earth system in which both the dependent and independent variables of interest are themselves uncertain[32,67].

We compare the model-predicted time series generated through this process with the observational SWE product on which the model is trained, using the common $R^2$ and RMSE metrics (Extended Data Fig. 4). In addition, as the emphasis of the analysis is on long-term trends in SWE, we compare the reconstructed trends with the observed trends over the study period and find that our models faithfully reproduce the spatial pattern and magnitude of the trends quite well, with correlations for all data products falling between 0.9 and 0.97 (Extended Data Fig. 3).

Furthermore, the RMSE of the construction model predictions is comparable across the 10 coldest, 10 warmest and 20 'average' years in the 1981–2020 period, indicating that the reconstructions are stable even in extreme years (Supplementary Fig. 4).

As an additional test of model skill, we use the model trained on only the gridded observational products to predict fully out-of-sample March SWE at 2,961 in situ sites from the SNOTEL, CanSWE and NH-SWE datasets. Our reconstructions are able to capture the interannual variability in in situ SWE quite well, with a median $R^2$ across stations of 0.59 and an RMSE of around 22% (Extended Data Fig. 5). The reconstruction model predictions are similarly able to capture skillfully the long-term SWE trends at the in situ sites, with a pattern correlation of 0.72 (Extended Data Fig. 5). Finally and crucially, we confirm that there are no systematic trends in time of the bias of our reconstructions against the in situ observations (Supplementary Fig. 5), indicating that the reconstruction models are capturing the real-world rate of change of snowpack with high fidelity.

## Counterfactual snowpack reconstructions

To identify where and how anthropogenic climate change has altered spring snowpack at impact-relevant scales, we combine our observation-based reconstructions, which are highly skilful at capturing historical SWE trends at impact-relevant scales, with climate model simulations that allow us to estimate forced changes to temperature and precipitation. Such a data–model fusion approach has been used to attribute anthropogenically forced changes to a wide variety of systems, both physical (for example, soil moisture[8,31], wildfire[30] and lake water storage[32]) and socioeconomic (for example, crop indemnities[33] and climate damages[34]).

We calculate the temperature response to anthropogenic forcing as the difference between the 30-year rolling mean average temperature for each month in the HIST and HIST-NAT runs. For precipitation, we calculate the forced response as the percentage difference between 30-year rolling mean monthly precipitation in HIST versus HIST-NAT. By differencing experiments from the same model, we hope to limit the influence of model biases in climatological temperature and precipitation, as each model is benchmarked to its own climatology. Systematic biases in the model-simulated trends (for example, too rapid warming or wetting), however, could potentially lead to over- or under-estimating the forced response. To address this possibility, we evaluate model biases in the 1981–2020 trends in winter temperature and precipitation against observed trends by taking the difference between the CMIP6 HIST ensemble mean and the mean of the observational products for each quantity (Extended Data Fig. 6). To test whether the observed and modelled trends are consistent, we ask whether the observed trend falls within a plausible range of forcing plus internal variability, given as the 2.5–97.5th percentile range of the CMIP6 HIST trends. Only 1% (3%) of grid cells fall outside this range for temperature (precipitation), indicating that the climate models capture realistic historical climate trends.

Having estimated anthropogenically forced changes in gridded temperature and precipitation, we create counterfactual time series of temperature and precipitation by downscaling the output to the $0.5° \times 0.5°$ resolution of the observational ensemble using conservative regridding and removing the forced response from each model realization from each gridded temperature and precipitation dataset. Temperature is adjusted by subtracting the forced change from the observations and precipitation is adjusted by the forced percentage change. Then, we use the reconstruction models trained on historical data (equation (1)) to predict March SWE using the counterfactual temperature and precipitation data, giving an estimate of what SWE would have been absent human-caused climate change. In addition, we isolate the effects of forced changes to temperature and precipitation individually by removing the forced response of only one or the other quantity from the observations, while leaving the other at its observed

historical values. These gridded counterfactual reconstructions are then similarly aggregated to the basin scale and linear trends in SWE for these counterfactual scenarios are calculated using the Theil–Sen estimator. The effect of forced changes to temperature and precipitation individually (Fig. 3c,d) and in combination (Fig. 3e) is calculated as the difference between each historical trend and the counterfactual trends based on the same SWE–temperature–precipitation dataset combination. For each of the 120 reconstruction ensemble members, we have 101 estimates of the anthropogenic effect (one from each climate model realization; Extended Data Table 2), for a total of 12,120 estimates for each basin. Using only the first realization from each climate model, rather than all available runs, produces nearly identical results (Supplementary Fig. 6).

To further test the validity of this approach of using forced changes in temperature and precipitation to estimate counterfactual SWE, we repeat this protocol using exclusively climate model output in a 'perfect model' framework. For each model, we fit the empirical model described in equation (1) using SWE, temperature and precipitation data from the CMIP6 HIST simulations over the 1981–2020 period, rather than observations. Then, we use the random forest trained on these HIST data to predict counterfactual SWE using temperature and precipitation from the HIST-NAT simulations. Finally, we compare the forced (HIST minus HIST-NAT) trends calculated from the reconstruction approach to the 'true' forced trends calculated by using the direct SWE output from the HIST and HIST-NAT climate model experiments (Extended Data Fig. 9 and Supplementary Fig. 7). The strong similarity in the patterns of the 'true' and reconstructed forced responses indicates that using observations with forced changes in temperature and precipitation removed produces reasonable estimates of a forced SWE change.

## Uncertainty quantification

The methods detailed above yield 12,120 estimates of the effect of climate change on March snowpack trends in each of 169 major river basins. Contributing to the spread of these estimates are four main sources of uncertainty: (1) uncertainty in the SWE data products on which the reconstructions are based; (2) uncertainty in the temperature and precipitation data products and their relationship with SWE; (3) differences in the forced response of temperature and precipitation due to structural differences between climate models; and (4) uncertainty due to internal climate variability in temperature and precipitation.

To quantify the magnitude of uncertainty introduced by each source, we calculate the standard deviation of forced SWE trends across a single dimension, holding all others at their mean. For instance, the uncertainty due to differences in model structure is given by the standard deviation of forced SWE trends across the 12 climate models (considering only the first realization from each), taking the mean across all SWE–temperature–precipitation dataset combinations.

To isolate the uncertainty from internal variability in temperature and precipitation, we use 50 pairs of HIST and HIST-NAT simulations from the MIROC6 model[68], which differ in only their initial conditions. We take the standard deviation of forced SWE trends for all 50 realizations, taking the mean across all SWE, temperature and precipitation data product combinations.

Consistent with previous work in uncertainty partitioning[19,41,69], we consider total uncertainty $U$ in the forced SWE trend in basin $b$ to be the sum of all four sources:

$$U_b = S_b + TP_b + M_b + I_b \qquad (2)$$

where $S$ is the uncertainty from SWE observations, TP is the uncertainty from temperature and precipitation observations, $M$ is the uncertainty from model structure, and $I$ is the uncertainty from internal variability. To assess which sources are the largest contributor to uncertainty

in each basin, we consider the fractional uncertainty of each (for example, $S_b/U_b$ gives the proportion of uncertainty in basin $b$ attribution to SWE observational uncertainty). This fractional uncertainty is reported in Supplementary Fig. 12. For each source, we hatch out basins where the magnitude of uncertainty is insufficient to change the sign of the ensemble mean estimate of the forced SWE trend (that is, the signal-to-noise ratio is >1).

## Temperature sensitivity of snowpack

To better understand the drivers of the heterogeneous spatial response of SWE and its potential future changes with further warming, we evaluate the temperature sensitivity of March SWE across a gradient of climatological winter temperatures in in situ observations, gridded observations, our basin-scale reconstructions and climate models. The marginal effect of an additional degree of warming, $\partial SWE/\partial T$ or $\beta_1$, is calculated as the regression coefficient of March SWE on cold-season (November–March) temperature:

$$SWE_{y,i} = \beta_{0,i} + \beta_{1,i} T_{y,i} \qquad (3)$$

where $SWE_{y,i}$ is March SWE in unit $i$ (in situ station, grid cell or river basin) in water year $y$ and $T_{y,i}$ is average cold-season temperature in that same unit. We run this regression at each in situ location, for all 20 combinations of gridded SWE and temperature products, for all 12 climate models (using the HIST simulations), and for all 120 basin-scale reconstructions. We then calculate the average and standard deviation of all of the coefficients for a given type of data (in situ, gridded observations, climate models and basin-scale reconstructions) in a rolling 5° temperature window to produce the curves in Fig. 4a. As such, the uncertainty estimate includes both parametric and data uncertainty.

## Snowpack-driven runoff changes

To evaluate the differential water security implications of the human-caused snowpack declines, we quantify the spring (April–July) runoff change due to forced March SWE changes. We once again use the random forest algorithm, modelling April–July run-off as a function of March SWE and monthly temperature and precipitation from the previous November to July:

$$Q_{y,b} = f(SWE_{y,b}, T_{y,11,b}, P_{y,11,b}, T_{y,12,b}, P_{y,12,b}, \ldots, T_{y,7,b}, P_{y,7,b}) \qquad (4)$$

where $Q_{y,b}$ is April–July total runoff in water year (October–September) $y$ in basin $b$, $SWE_{y,b}$ is average March SWE in water year $y$ in basin $b$—unlike the SWE reconstructions, which were fit at the grid-cell level and aggregated to the basin scale, the runoff model is fit using basin-scale data—$T_{y,m,b}$ is the area-weighted basin-average temperature in month $m$ of water year $y$, and $P_{y,m,b}$ is the total basin-scale precipitation in month $m$ of water year $y$. We fit this model using all 120 SWE–temperature–precipitation dataset combinations and the GloFAS runoff data (Extended Data Table 1). We evaluate model skill using the same methods as those used to validate our SWE reconstructions (Extended Data Fig. 10).

Analogous to the basin-scale March SWE attribution described above, the spring runoff change due to forced changes to snowpack is given by the difference between runoff estimated with historical SWE and runoff estimated with the effects of forced temperature and precipitation changes on SWE removed.

## Future snowpack and runoff changes

To better understand the differential water-availability implications of future warming-driven SWE changes, we combine our statistical models and projections of future temperature and precipitation change to produce estimates of end-of-century (2070–2099) snowpack under the SSP2-4.5 forcing scenario. Specifically, we use a 'delta' method in which we adjust the observed climatology for each month by the

difference between the end-of-century and historical (1981–2020) climate from the climate models. We additively adjust temperature and adjust precipitation by the percentage change between historical and future climate. We then make predictions of future climatological snowpack using the adjusted data and the model described in equation (1) trained on historical data.

Future runoff changes due to changes in SWE are calculated using equation (4), but substituting estimates of future SWE climatology for the historical, while keeping temperature and precipitation at their observed historical climatological values.

### Snow dominance

To identify a priori the river basins considered to be snow dominant in Fig. 1, we use the ratio $R$ of water year (October–September) cumulative snowfall to runoff[1], calculated from ERA5-Land[45]. Basins where the average $R$ is greater than 0.5 are considered to be snowmelt dominant.

### Data availability

All data that support this study are publicly available at the following locations: CMIP6 model outputs, https://esgf-node.llnl.gov/; SNOTEL, https://wcc.sc.egov.usda.gov/nwcc/tabget; CanSWE, https://zenodo.org/records/5889352; NH-SWE, https://zenodo.org/records/7565252; Snow-CCI, https://climate.esa.int/en/projects/snow/Snow_data/; ERA5, ERA5-Land and GloFAS, https://cds.climate.copernicus.eu/. JRA-55, https://rda.ucar.edu/datasets/ds628.0/; MERRA-2, https://gmao.gsfc.nasa.gov/reanalysis/MERRA-2/; TerraClimate, https://www.climatologylab.org/terraclimate.html; GPCC, https://psl.noaa.gov/data/gridded/data.gpcc.html; MSWEPv280, http://www.gloh2o.org/mswep/; Berkeley Earth, https://berkeleyearth.org/data/; Climate Prediction Center (CPC), https://www.cpc.ncep.noaa.gov/; Gridded Population of the World (GPW), https://sedac.ciesin.columbia.edu/data/collection/gpw-v4; Global Runoff Data Center Major River Basins, https://www.bafg.de/GRDC/. Source data are provided with this paper.

### Code availability

All code that supports this study is available at https://doi.org/10.5281/zenodo.10035276.

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

**Acknowledgements** We thank Dartmouth's Research Computing and the Discovery Cluster for computing resources; the World Climate Research Programme, which, through its Working Group on Coupled Modeling, coordinated and promoted CMIP6; and all climate modelling groups for producing and making available their model output. We acknowledge funding for this research from NOAA MAPP NA20OAR4310425 (J.S.M.) and DOE DESC0022302 (J.S.M. and A.R.G.).

**Author contributions** Both authors designed the analysis. A.R.G. performed the analysis. Both authors interpreted the results and wrote the paper. J.S.M. funded the research.

**Competing interests** The authors declare no competing interests.

**Additional information**
**Correspondence and requests for materials** should be addressed to Alexander R. Gottlieb.

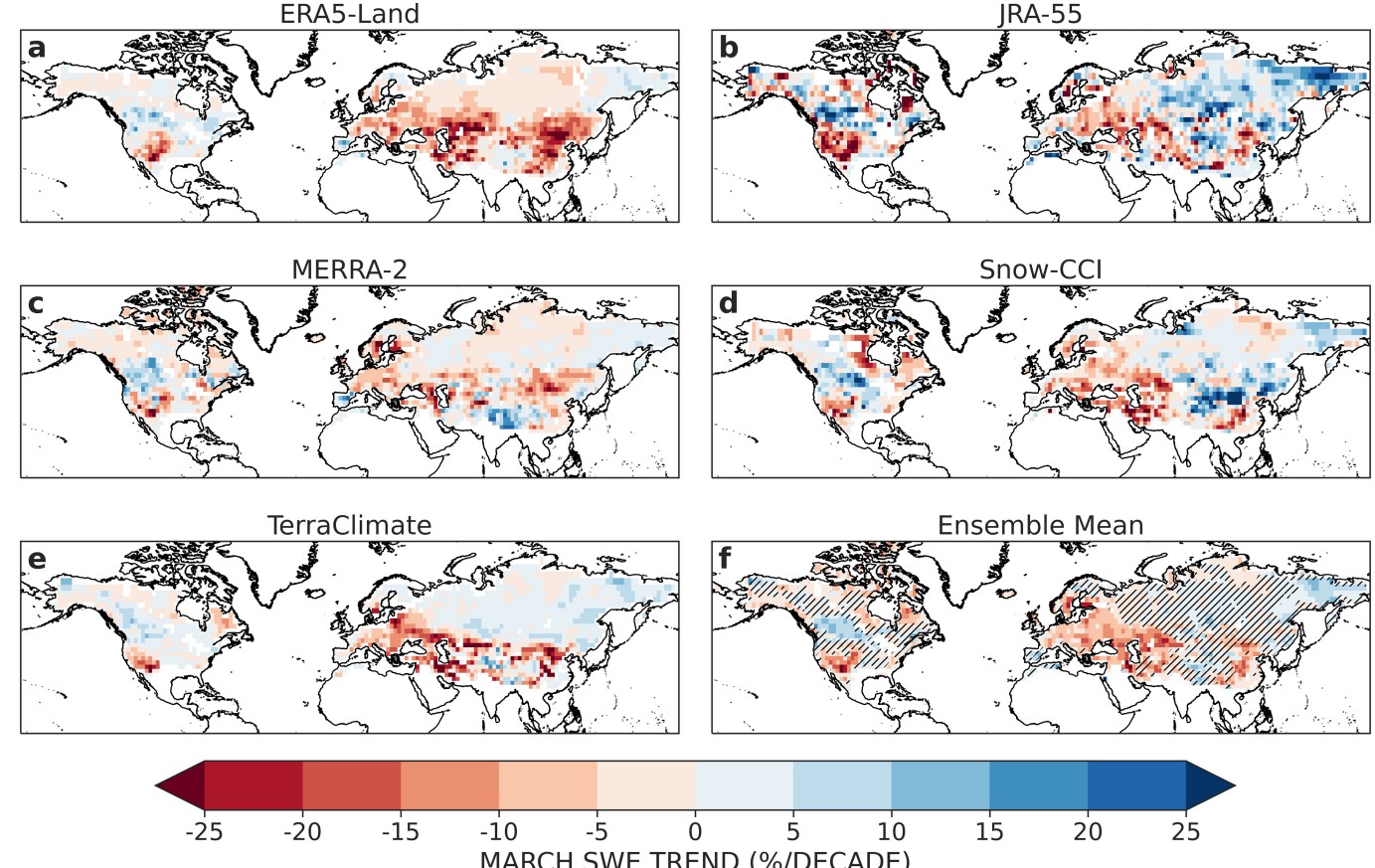

**Extended Data Fig. 1 | Heterogenous long-term trends in observed March SWE make claims about snow responses to warming a challenge. a-e**, Trend in March SWE from 1981 to 2020 from individual gridded SWE data products. **f**, Average trend across all 5 products. Grid cells where fewer than 4 products agree on the sign of the trend are hatched. Maps were generated using cartopy v0.18.0.

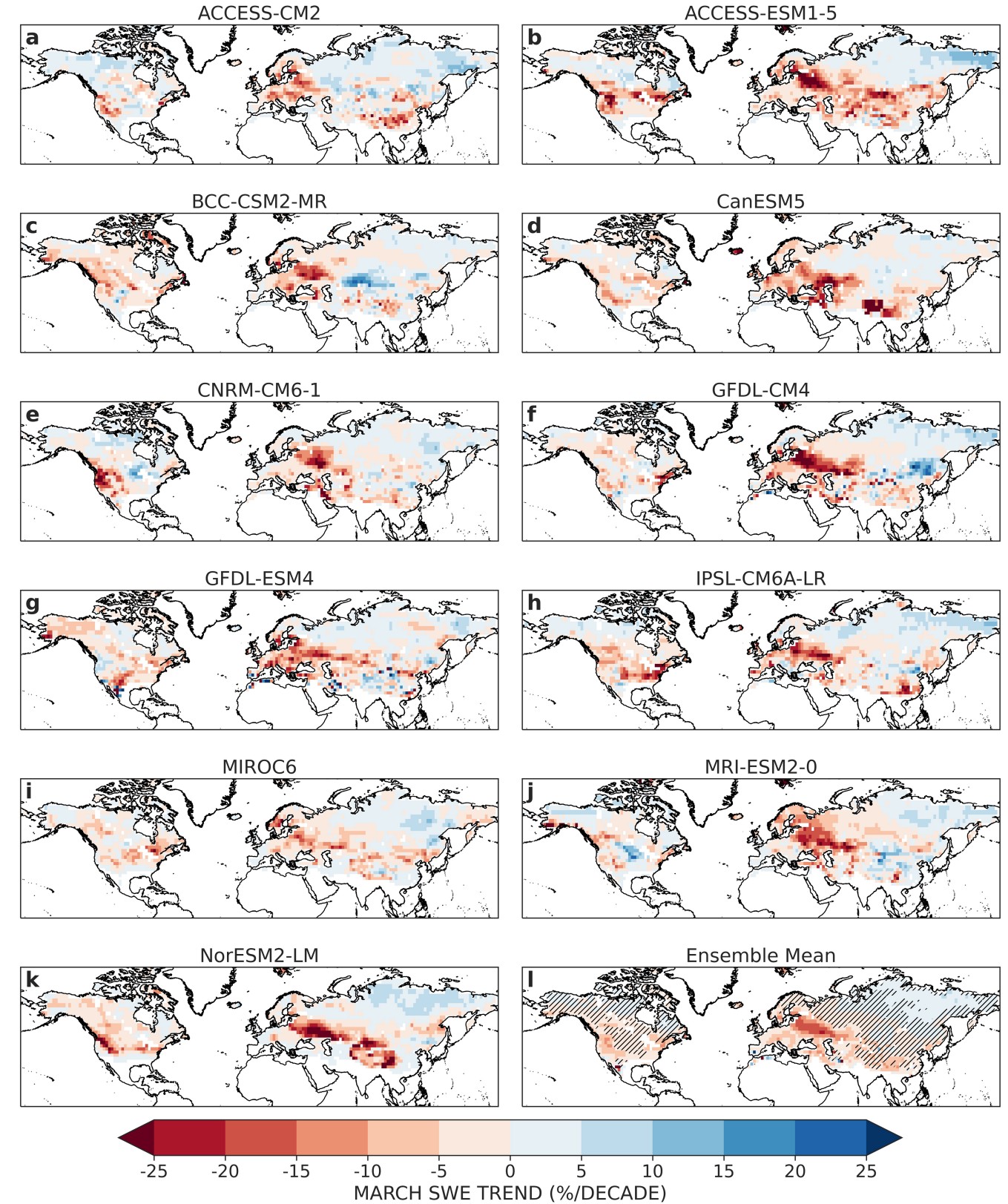

**Extended Data Fig. 2 | Historical trends in March SWE from CMIP6 models exhibit uncertainty outside of the Western United States, Europe, and Northern Eurasia. a-k**, Trend in March SWE from 1981 to 2020 from historical climate model simulations. Details of models can be found in Extended Data Table 2. **l**, Ensemble mean trend. Grid cells where fewer than 80% of models agree on the sign of the trend are hatched. Maps were generated using cartopy v0.18.0.

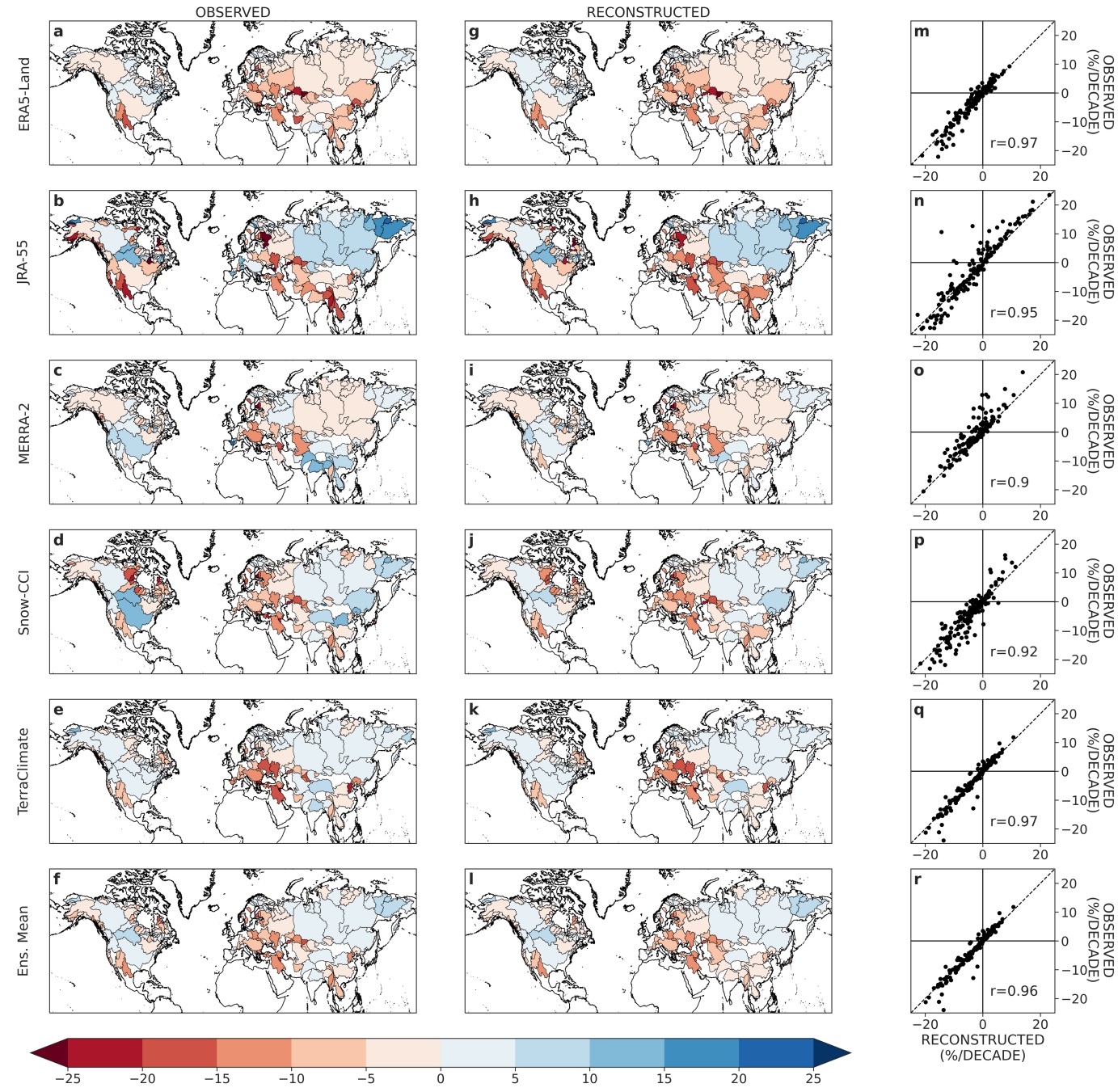

**Extended Data Fig. 3 | Ensemble reconstructions based on the Random Forest model skillfully reproduce the pattern and magnitude of long-term SWE trends in each snow product.** Observed (**a**-**f**) and reconstructed (**g**-**l**) 1981–2020 March SWE trends for 5 gridded SWE data products and their mean. **m**-**r**, Scatterplot of reconstructed versus observed trends, where each dot represents a river basin. Dashed line denotes perfect reconstruction. Pearson's correlation is shown in bottom right corner. Maps were generated using cartopy v0.18.0. River basin boundaries come from the Global Runoff Data Centre's Major River Basins of the World database[44].

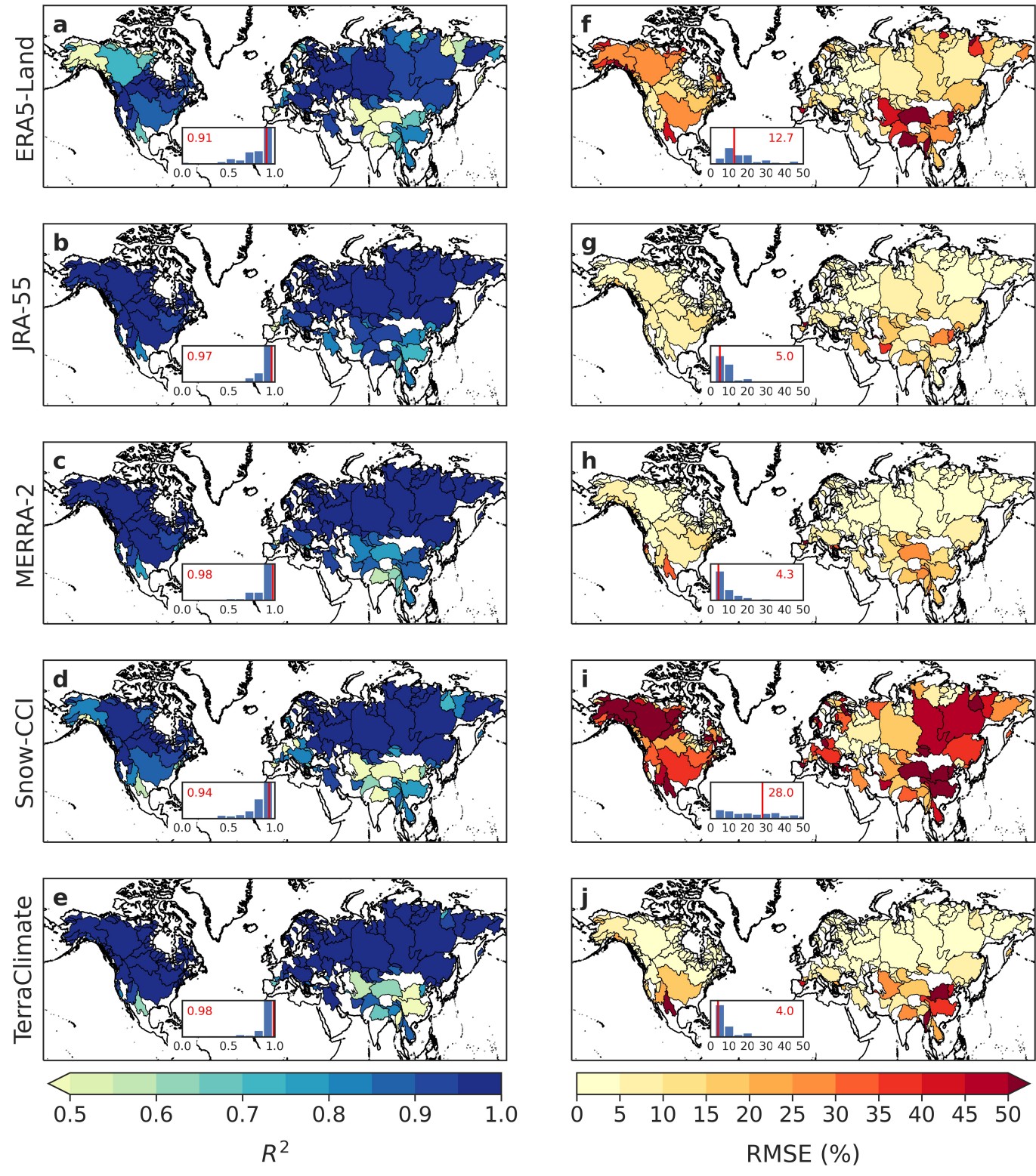

**Extended Data Fig. 4 | The Random Forest model exhibits high snowpack reconstruction skill based on temperature and precipitation data.** Basin-scale R² (**a-e**) and root-mean-square error (RMSE; **f-j**) for 5 gridded SWE data products over the period 1981–2020. Each metric shows the skill of the mean of all reconstructions for a single SWE product versus the observed values from that product. Insets show the distribution of skill across basins, with the red line and value indicating the median. Maps were generated using cartopy v0.18.0. River basin boundaries come from the Global Runoff Data Centre's Major River Basins of the World database[44].

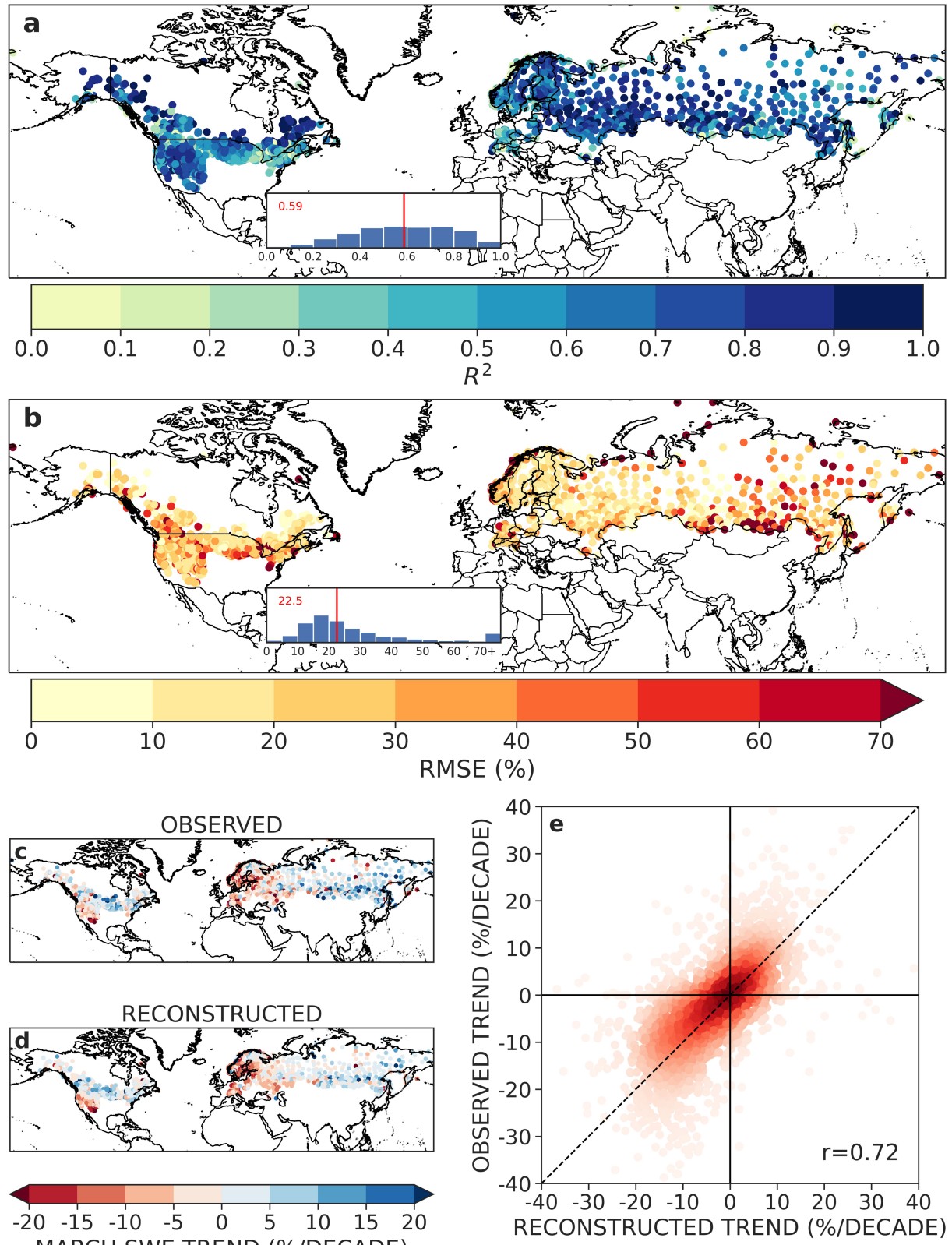

**Extended Data Fig. 5 | The ensemble reconstruction based on the Random Forest model skillfully predicts the variability and trends in out-of-sample** *in situ* **snowpack data.** $R^2$ (**a**) and RMSE (**b**) of Random Forest model predictions of *in situ* March SWE at 2,961 locations over the period 1981–2020. Insets show the distribution of skill across sites, with the red line and value indicating the median. Observed (**c**) and reconstructed (**d**) 1981–2020 March SWE trends.

**c**, Scatterplot of reconstructed versus observed trends, where each dot represents an *in situ* location. Points are colored by their density. Dashed line denotes perfect agreement between reconstructed and observed trends. Pearson's correlation is shown in bottom right corner. Maps were generated using cartopy v0.18.0.

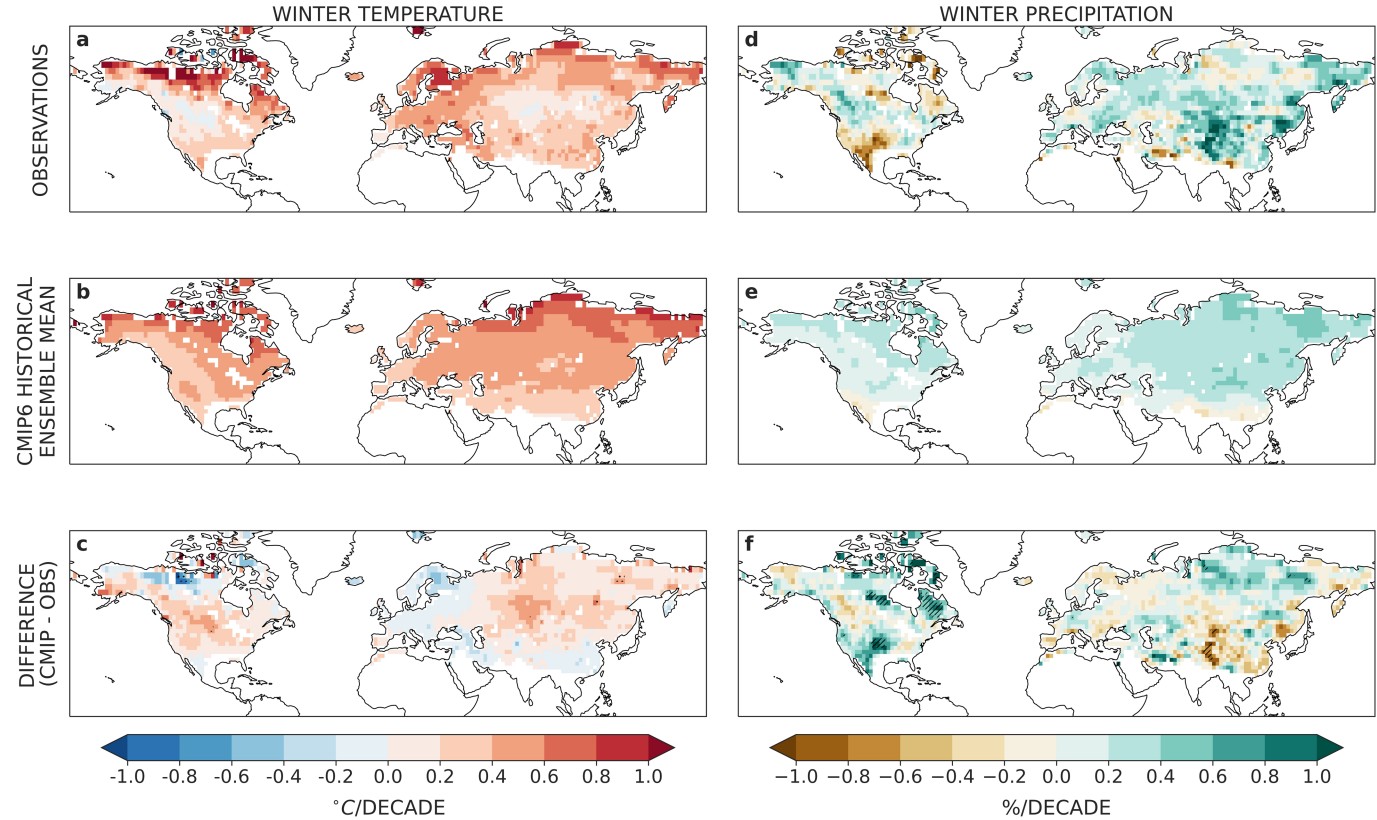

**Extended Data Fig. 6 | CMIP6 model bias in winter temperature and precipitation trends largely within range of natural variability.** Observed trends in November-March average temperature (**a**) and total precipitation (**d**) from 1981 to 2020. **b**, **e**. Ensemble mean of historical CMIP6 simulations. **c**, **f**. Average bias in trends across all observation-model combinations. Hatching indicates regions where the observed trend falls outside the 2.5–97.5th percentile range of the CMIP6 trends. Maps were generated using cartopy v0.18.0.

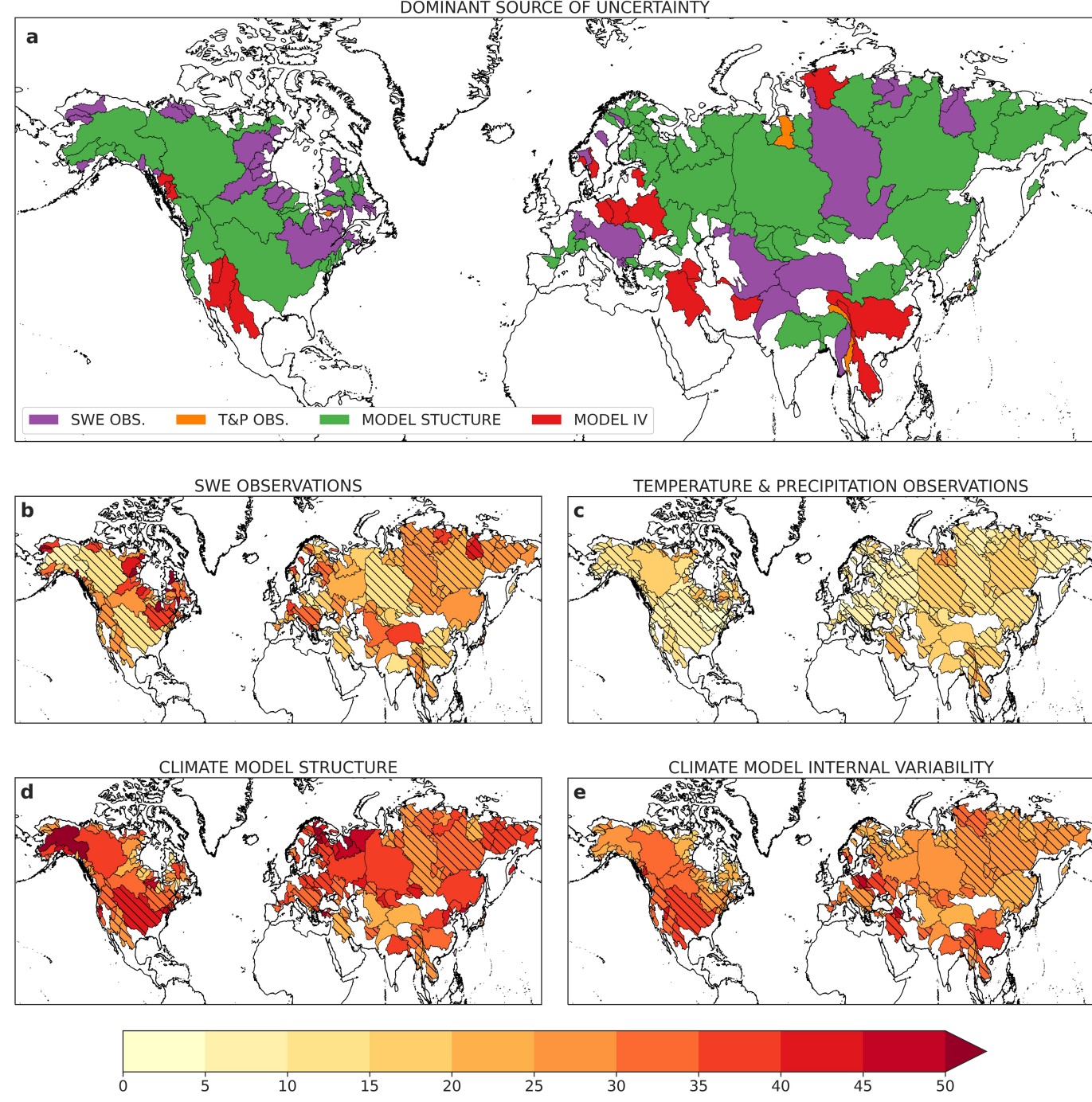

**Extended Data Fig. 7 | Uncertainty in the attribution of human-caused snowpack trends resides with climate model structure and modeled internal variability, not observations. a**, Dominant source of uncertainty in reconstruction-based estimates of forced March SWE trends from 1981 to 2020. **b-e**, Percentage of total uncertainty in forced SWE trends attributable to (**b**) observational uncertainty in gridded SWE products, (**c**) observational uncertainty in temperature and precipitation data products, (**d**) uncertainty in the forced response of temperature and precipitation across different climate models, and (**e**) uncertainty in the forced response of temperature and precipitation arising from internal variability (Methods). Hatching indicates basins where the uncertainty attributable to a given source is insufficient to change the sign of the ensemble mean estimate of the forced SWE trend. Maps were generated using cartopy v0.18.0. River basin boundaries come from the Global Runoff Data Centre's Major River Basins of the World database[44].

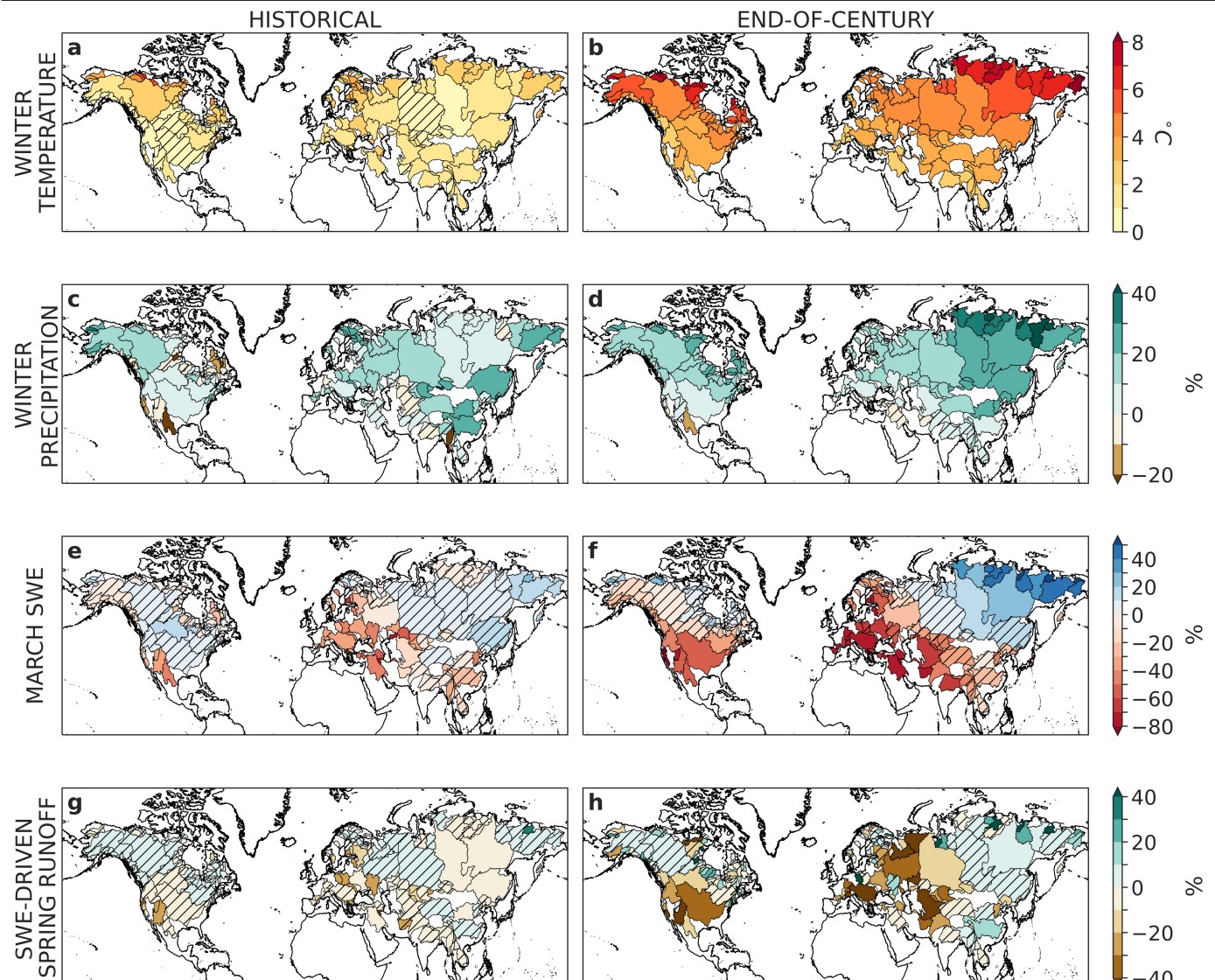

**Extended Data Fig. 8 | Historical associations among climate, snowpack, and snow-driven runoff portend accelerating changes to snow hydrology.** Column 1: Historical change in November-March average temperature (**a**), total precipitation (**c**), March average SWE (**e**), and snowpack-driven April-July runoff (**g**) over the period 1981–2020. Values represent averages across all data products and hatching indicated basins where fewer than 80% of products agree on the sign of the change. Column 2: 2070–2099 changes under the SSP2-4.5 forcing scenario relative 1981–2020. Temperature (**b**) and precipitation (**d**) are calculated as the difference within each model realization between the end-of-century and climatological periods and future SWE (**f**) and runoff (**h**) changes are calculated according to Equations 1 and 2, respectively. Maps were generated using cartopy v0.18.0. River basin boundaries come from the Global Runoff Data Centre's Major River Basins of the World database[44].

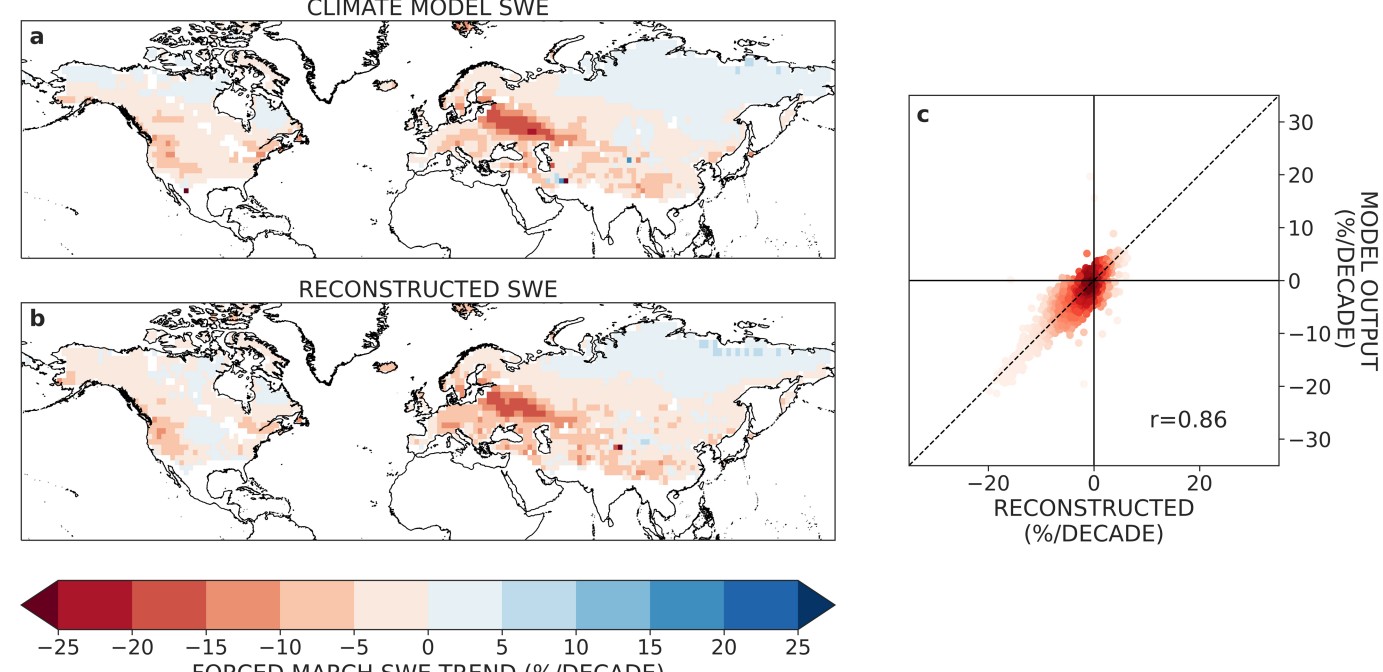

**Extended Data Fig. 9 | The Random Forest snowpack reconstruction methodology exhibits high skill based on a perfect model framework.** CMIP6 ensemble mean forced (HIST minus HIST-NAT) trends in March SWE from 1981–2020 based on (**a**) climate model SWE output and (**b**) SWE estimated using climate model temperature and precipitation and Random Forest model. **c**, Scatterplot of reconstructed versus original trends, where each dot represents a grid cell. Points are colored by their density. Dashed line denotes perfect agreement between reconstructed and original trends. Spatial correlation is shown in the bottom right corner.

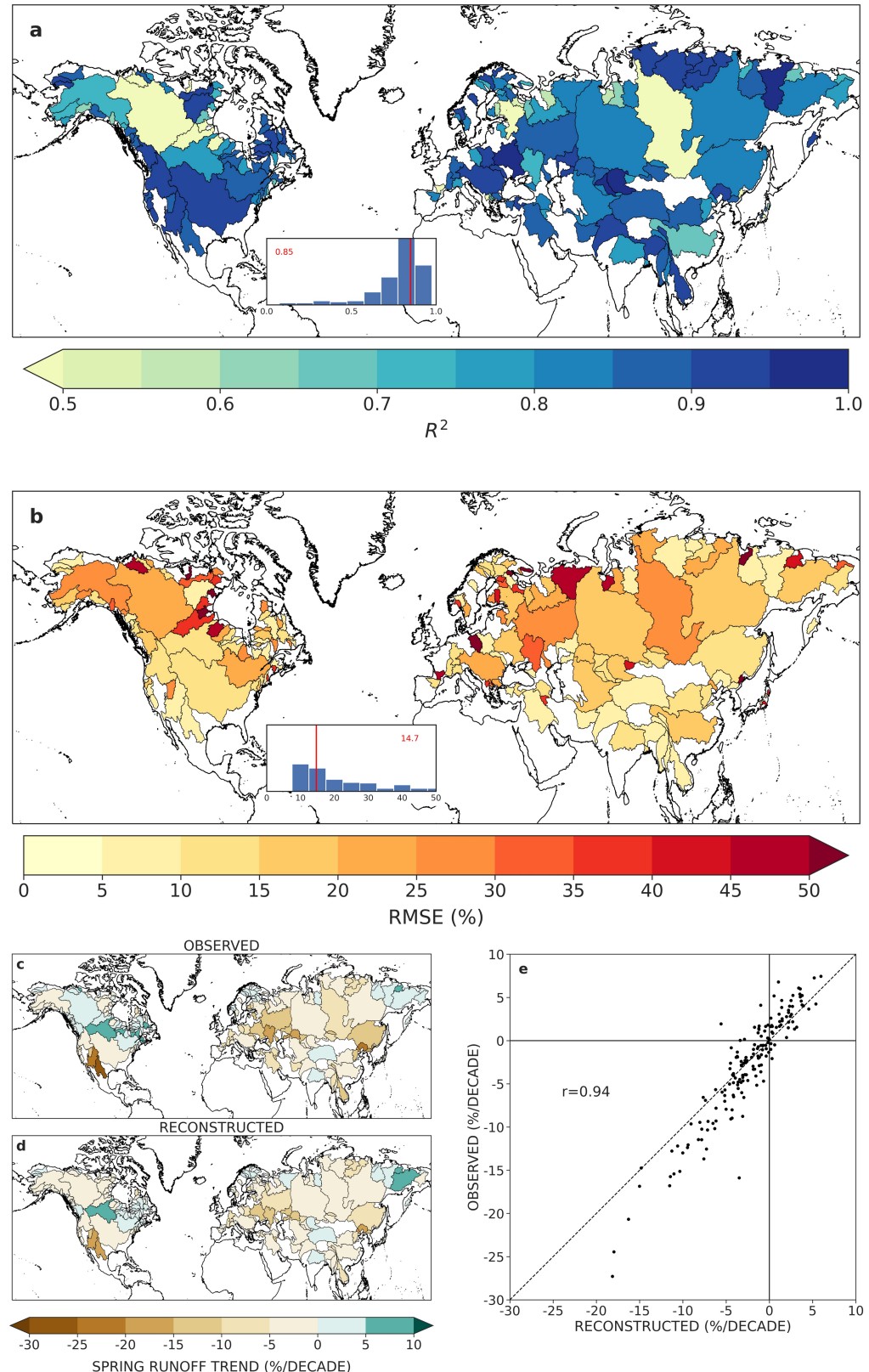

**Extended Data Fig. 10 | The Random Forest model is extended to predict runoff from snowmelt skillfully.** $R^2$ (**a**) and RMSE (**b**) of Random Forest model predictions of April-July basin-scale runoff from 1981 to 2020. Insets show the distribution of skill across sites, with the red line and value indicating the median. Observed (**c**) and reconstructed ensemble mean (**d**) 1981–2020 April-July runoff trends. **e**, Scatterplot of reconstructed versus observed trends, where each dot represents a basin. Dashed line denotes perfect agreement between reconstructed and observed trends. Spatial correlation is shown in center left. Maps were generated using cartopy v0.18.0. River basin boundaries come from the Global Runoff Data Centre's Major River Basins of the World database[44].

**Extended Data Table 1 | Summary of observational data products used in the analysis**

| Dataset | Dataset Type | Resolution | Years |
|---|---|---|---|
| **Snow Water Equivalent** | | | |
| SNOTEL | In situ | Point (N=550) | 1977-present |
| CanSWE | In situ | Point (N=341) | 1928-2020 |
| NH-SWE | In situ (snow depth converted to SWE) | Point (N=2119) | 1950-2020 |
| ERA5-Land* | Reanalysis | 0.1° x 0.1° | 1950-present |
| JRA-55* | Reanalysis | 55km x 55km | 1958-present |
| MERRA-2* | Reanalysis | 0.5° x 0.625° | 1981-present |
| Snow-CCI | Passive remote sensing + in situ | 25km x 25km | 1981-present |
| TerraClimate* | Observed T&P + water balance model | 0.04° x 0.04° | 1958-present |
| **Precipitation** | | | |
| ERA5 | Reanalysis | 0.25° x 0.25° | 1980-present |
| GPCC | Interpolated gauge | 0.5° x 0.5° | 1891-present |
| MERRA-2 | Reanalysis | 0.5° x 0.625° | 1981-present |
| MSWEP | Merged satellite/gauge | 0.1° x 0.1° | 1979-present |
| TerraClimate | Merged gauge/reanalysis | 0.04° x 0.04° | 1958-present |
| **Temperature** | | | |
| Berkeley Earth (BEST) | Interpolated in situ | 1° x 1° | 1753-present |
| Climate Prediction Center (CPC) | Interpolated in situ | 0.5° x 0.5° | 1979-present |
| ERA5 | Reanalysis | 0.25° x 0.25° | 1979-present |
| MERRA-2 | Reanalysis | 0.5° x 0.625° | 1981-present |

*indicates SWE product used in Fig. 1.

**Extended Data Table 2 | Summary of CMIP6 models used in the analysis**

| Model Name | Modeling Center | Resolution | Realizations (#) |
|---|---|---|---|
| ACCESS-CM2 | Commonwealth Scientific and Industrial Research Organisation | 1.875°×1.25° | r[1-3]i1p1f1 (3) |
| ACCESS-ESM1-5 | Commonwealth Scientific and Industrial Research Organisation | 1.875°×1.25° | r[1-3]i1p1f1 (3) |
| BCC-CSM2-MR | Beijing Climate Center | 1.1°×1.1° | r1i1p1f1 (1) |
| CNRM-CM6-1 | Centre National de Recherches Météorologiques/ Centre Européen de Recherche et de Formation Avancée en Calcul Scientifique | 1.4°×1.4° | r1i1p1f2 (1) |
| CanESM5 | Canadian Centre for Climate Modelling and Analysis | 2.8°x2.8° | r[1-25]i1p1f1 (25) |
| FGOALS-g3* | State Key Laboratory for Numerical Modeling for Atmospheric Science and Geophysical Fluid Dynamics | 2°x2° | r1i1p1f1 (1) |
| GFDL-CM4 | Geophysical Fluid Dynamics Laboratory | 1°x1° | r1[i1p1f1 (1) |
| GFDL-ESM4 | Geophysical Fluid Dynamics Laboratory | 1°x1° | r[1-3]i1p1f1 (1) |
| IPSL-CM6A-LR | Institut Pierre-Simon Laplace | 1.27°x2.5° | r[1-2,4-6]i1p1f1 (5) |
| MIROC6 | International Centre for Earth Simulation | 1.4°x1.4° | r[1-50]i1p1f1 (50) |
| MRI-ESM2-0 | Meteorological Research Institute | 1.1°x1.1° | r[1-5]i1p1f1 (5) |
| NorESM2-LM | Norwegian Climate Center | 2°x2° | r[1-3]i1p1f1 (3) |

*No monthly snow water equivalent from the pre-industrial control run archived.

Monthly snow water equivalent "snw" is used from the "piControl", "historical" (1850–2015), "historical-nat" (1850–2020) and "ssp245" (2015–2100) experiments. Monthly air temperature ("tas") and precipitation ("pr") is used from the "historical", "historical-nat", and "ssp245" experiments. *No monthly snow water equivalent from the pre-industrial control run archived.