## [Peer Review File · Nature]

Manuscript Title: Evidence of human influence on Northern Hemisphere snow loss

Reviewer Comments & Author Rebuttals

Reviewer Reports on the Initial Version:

Referees' comments:

Referee #1 (Remarks to the Author):

The paper shows an interesting analysis of the effects of anthropogenic climate change on snowpack and, consequently, runoff in boreal regions. In general, I found the approach novel, providing new important results that are worth publishing in Nature. The assessment of human influence on snow cover evolution is carried out by using an innovative approach that is based on a training of an empirical model. Overall, the analysis is welcomed. The paper is also well written.

However, there are some issues affecting the reliability of presented results, thus requiring further consideration. A major issue is that the used observational data on snow water equivalent (SWE) is not really observational, but instead model predictions incorporating snow depth observations as model input. For example, the applied Canadian Meteorological Centre's Daily Snow Depth Analysis uses synoptic snow depth observations from weather stations as input to a model that predicts snow accumulation and melt (Brown & Brasnett, 2010). The more complex reanalysis models, also used here, do basically the same. But when snow depth observations are assimilated to models, problematic issues include how well is the snow density evolution considered when assessing the magnitude of SWE. As an outcome the ensemble of model predictions used here as 'observational' data set representing the peak SWE (1 April) may include systematic errors that seriously disturb the reliability of regional trends, see e.g. (Pulliainen et al. 2020) that discusses the reliability of SWE trends among different snow products. To consider this issue the authors should investigate and compare their data with the available SWE information including available historical distributed SWE observation data sets made in snow courses across Eurasia and Canada (see e.g., NSDC data archives). This comparison would evidently make the uncertainty analyses of the paper more convincing. Currently, the methods part of the paper indicates (L.346) that the SNOTEL network in the Western U.S. is used for the validation of the SWE reconstruction. This is not sufficient for the hemispheric analysis as the single point snow pillow measurements of SWE in the SNOTEL network are limited to mountainous regions of the Western U.S.

Detailed line comments:

L.71-73: The possible change of bias with time in SWE products is an additional source of error that is not considered.

L.83: References 25 and 26 only consider mountain regions (Western U.S. and European Alps).

L.363: The SWE model used for reconstructing the historical April 1 SWE is simple, but robust and useful for the presented analysis.

L.368-370: The sentence ignores the limitations, as the paper does not currently employ the in situ SWE observations distributed across the Northern Hemisphere. References included here (25, 26, 59) only deal with regional analysis in some mountain regions (European Alps and Western U.S.) .

L.383: Results of Fig. S4 indeed capture the interannual variability of SWE quite well at the SNOTEL sites of the Western U.S. for the period 2000-2020 (i.e., 21 samples for each site). However, apparently any conclusion on trends cannot be made (or they are not shown)? Same kind of analysis could be made across the hemispheric scale using in situ snow course data (requires some interpolation as observations are typically available on a monthly, bi-monthly, or weekly basis for a given snow course).

L.421: SWE-T-P evidently refers to temperature and precipitation, text would be less confusing (in general) if the number of abbreviations were reduced. Also e.g., abbreviation "NoACC" is somewhat problematic even though explained for the first time in L.99.

L.431: Uncertainty in SWE data products is not adequately considered concerning the possible temporally changing bias of these products, see the general comment above.

L.478: Since Eq. (6) is an implication of Eqs. (4) and (5) it could be expressed/clarified.

L.498: It is unclear how the $dSWE/dT$ curve is determined, as Eq. (7) is evidently providing a constant value for each basin?

Figures 1 and 2: Obtained numbers for the major basins/ivers discharging to the Arctic Ocean would be interesting to see as well (even though they represent sparsely populated areas).

Figure S8: Reader can be confused with the multitude of abbreviations, such as AMJJ.

References:

Brown, R. & Brasnett, B. Canadian Meteorological Centre (CMC) Daily Snow Depth Analysis Data, Version 1. NASA National Snow and Ice Data Center Distributed Active Archive Center. (2010) doi:10.5067/W9FOYWH0EQZ3.

Pulliainen, J. et al. Patterns and trends of Northern Hemisphere snow mass from 1980 to 2018. *Nature*, 581:294-298 (2020).

Reviewed by Jouni Pulliainen

Referee #2 (Remarks to the Author):

This study, by Gottlieb and Mankin, uses an ensemble of snow products and climate models to investigate climate change impacts on basin-scale snowpack and runoff in the northern hemisphere. The authors focus on April 1 SWE. They conclude that warming has reduced snow in the more highly populated basins and that future warming will have a worsening effect on SWE and runoff losses.

Overall, this study provides a nice dataset to investigate processes related to climate change. To me, it seems like an extension to the Mankin et al., 2015 paper (DOI 10.1088/1748-9326/10/11/114016) with similar conclusions. I like the attention to dataset uncertainties, but the manuscript tries to focus more on the runoff changes.

The methodology is OK, but the author's argue that snowpack assessments are quite sensitive to the data products analyzed and show disagreement between the products. However, they choose to use an ensemble approach with these products. My question is: why should we believe that the ensemble approach is better? Any mean value will be driven by the datasets that are further from

the truth (especially with only a handful of datasets). Furthermore, the largest RMSE values compared to SNOTEL (50% or greater, Fig. S4) end up being in the southwestern US in which much of the discussion is focused. Thus, I am not convinced that this ensemble is appropriately representing the analysis that the authors are attempting.

Additionally, the authors reconstruct snowpack as a function of temperature and precip., with temperature driving much of the changes. However, a recent paper has shown that temperature thresholds alone give large errors in precipitation phase estimates for the northern hemisphere, thus adding uncertainty to some of the main points about precipitation phase near a melting point threshold.

Reference: Jennings et al., 2018: <https://www.nature.com/articles/s41467-018-03629-7>

I would like to emphasize that I like the general approach in this study and that these comments are meant to be constructive. However, some improvements based on my comments above/below are required for publication, in my opinion.

Other minor comments:

the term "boreal": the term boreal is used in the title and elsewhere, but I disagree with it's use. In the snow hydrology community, boreal areas are generally between 50 and 60 or 65 degrees latitude whereas the authors appear to use it as everything south of 60 degrees.

line 31-33: if the data products are sensitive to the scale and analysis performed, what datasets may not be appropriate for the scale and analysis that was performed here?

lines 40-42: These are also the basins with some of the highest RMSE.

line 65: this statement "snowpacks across the Northern Hemisphere" is different than what the title implies.

Line 70: Not sure that I am buying it with that many large RMSE values.

line 151 (and elsewhere): please refrain from using terms like "very" as these are not well defined metrics (i.e., how cold is very cold?).

line 199-200: this sentence is a bit distracting for the conclusions section, in my opinion.

line 208-209: Please further expand on the quantifications of uncertainty sources. I gained little to nothing from the text about this and only a bit when looking at the figure, but nothing well quantified.

line 216-225: this is all discussion and nothing that resembles a conclusion. Please move to an appropriate section of the article.

Figures: Please change the projection. The current projection focuses on mid-latitudes and "squeezes" much of the northernmost basins.

Referee #3 (Remarks to the Author):

This study evaluates historical trends in basin-scale snowpack (SWE) and consequent snow-driven spring runoff over Northern Hemisphere during 1981-2020 using a statistical reconstruction approach. The authors argue that they have identified human-caused changes in SWE and consequent runoff ("attribution" in their definition) by comparing 40-year trends in the observation-based SWE reconstructions ("historical") with those in the counterfactual reconstructions without human influence ("NoACC").

I have focused on evaluating the attribution method as requested by the Editor. The attribution approach used in this paper has three steps:

First, the authors construct a multiple linear regression model (Eq. 1) of reconstructing (or predicting) April 1 SWE at each basin using cold season (NDJFM) temperature (T) and precipitation (P). They train this regression model using observational datasets during 2001-2020, where 8 SWE, 4 temperature (T), and 5 precipitation (P) datasets (including reanalyses and satellite measurements) are all combined to consider observational uncertainties, providing 160-member ensemble of observations and corresponding regression equations.

Second, for each observational-based member, counterfactual SWE reconstructions are estimated by inserting adjusted T and P into the pre-defined regression model. Here, T and P values are adjusted by removing possible anthropogenic changes from the observations. The anthropogenic changes in T and P are obtained from differences between historical (ALL) and historicalnat (NAT) simulations of available CMIP6 multi-model simulations (86 ALL-NAT values available).

Finally, the authors compare a distribution of 'historical' SWE trends (consisting of 160 observation-based estimates) with that of 'NoACC' SWE trends (consisting of 13,760 estimates obtained from 160 corresponding estimates \times 86 CMIP6 ALL/NAT runs). To claim that "historical SWE trends are attributable to human causes", the authors assess the significance of difference of the two SWE trend distributions using a Kolmogorov-Smirnov test (5% level) and also check the sign agreement of trend difference (ALL-NAT $>$ 0 in more than 75% of 86 runs) if my understanding is correct.

Applying this procedure to all snow-dependent river basins, the authors conclude that they can "confidently attribute changes in April 1 SWE to human causes in roughly 40 percent of the world's major river basins, including the 4 most populous snow-dominated basins in Europe and 3 of the top 4 in North America". They employ a similar statistical approach (Eq. 4) to further estimate snowpack-driven changes in spring runoff in the past and future conditions.

I find this study interesting in that it comprehensively takes account of observational uncertainties through SWE-T-P data combinations at the river basin scales (step 1 above). However, with regard to its attribution analysis (steps 2 and 3 above), I do not think that this approach can provide a way to claim "robust attribution of human-forced changes". Conventionally, attribution approaches quantify external forcing contributions to the 'observed' changes by either checking if the model-simulated forced pattern is present in the 'observations' (typically based on regressions but also using similarity measures) or comparing the likelihoods of the 'observed' changes between real-world and counterfactual conditions (similar to event attribution types of assessments). In contrast to the usual definition of attribution, this study evaluates human influences by comparing two estimates of reconstructed SWE trends, differences of which are largely determined by CMIP6-simulated T and P changes (ALL-NAT), irrespective of the significance and sign of 'observed' trends. In particular, cold season mean T differences between ALL and NAT (using their 30-year means) will induce SWE differences almost linearly based on the statistical relation in Eq. (1). In this way, resulting SWE differences likely reflect model response patterns, not indicating that this pattern is present in the observations or contributes to the observed changes. Since no observational information is used in this trend comparisons, my interpretation of their 'attribution' results would be such that CMIP6-simulated T and P responses to anthropogenic forcing are larger than observational uncertainties in T and P over many river basins or similar.

I have another concern about the stability of their statistical models. The statistical reconstruction method assumes that SWE dependence on T and P will not change with time but the validity of this

assumption is not discussed. Statistical models are trained using recent SWE, T, and P datasets for 2001-2020, from which regression coefficients (betas) are estimated. In this setting, SWE variations will be mainly associated with inter-annual variabilities in T and P during the short period. The authors, however, use the same betas to estimate observed SWE reconstructions for the early period for 1981-2000 as well as to estimate SWEs for much colder pre-industrial (for NoACC) and much warmer future conditions (+1 degree and even near the late 21st century). This extrapolation of statistical relation may induce large errors in SWE estimates. Same issue applies to spring runoff reconstructions in Eq. (4).

The other issue related to attribution is about the datasets used. Since many observational datasets are based on estimates from reanalyses and satellite measurements rather than real observations, it looks difficult to attribute historical changes. Regarding climate model simulations, the authors do not evaluate model performances at all. Depending on locations and seasons, models can systematically overestimate or underestimate T and P climatology and their responses to anthropogenic forcing. In this respect, Table S2 indicates that a single model provides 50 members out of total 86 members, potentially influencing results depending on the model's biases.

Overall, while detailed and useful information on snowpack and associated runoff changes due to warming is provided at river basin scales, I do not agree that this study provides robust attribution results due to the limitation of the statistical reconstruction approach.

Author Rebuttals to Initial Comments:

We thank the Reviewers for their thoughtful and constructive feedback, and the Editor for the helpful synthesis.

In response to the thoughtful comments from the Editor and Reviewers, we have made several important changes that greatly strengthen the methods and trustworthiness of the findings of the original submission of a detectable signal of human-caused warming on snowpack and snowmelt driven runoff. Briefly, these changes are as follows:

- 1. We now include far more *in situ* observational data (2,961 stations across the Northern Hemisphere versus 550 stations from only the Western U.S. in the original submission), which is now folded into each step of the analysis.**
- 2. We now employ an attribution at two spatial scales: hemispheric and river basin. The former uses a more traditional model attribution approach, showing that the observed pattern of SWE changes in both *in situ* observations and gridded products could not have emerged from internal variability alone. The latter clarifies the spatial pattern of these forced changes and their water availability implications.**
- 3. We now use a machine learning approach (Random Forest algorithm) to reconstruct SWE using all snow datasets, which we validate on the newly incorporated *in situ* data. Our revised empirical model markedly increases the skill of our reconstruction (median R^2 of >0.9 across all products and basins, versus 0.6 in the original) and positions our basin-scale attribution analysis.**

This Response to the Reviewers file provides a complete documentation of the changes that have been made in response to each individual Reviewer comment. This may lead to some redundancies where our changes are relevant to multiple Reviewer comments, but our aim is for the Editor and Reviewer to see how each comment was addressed, independent of any others.

Reviewer comments are shown in plain text. Author responses are shown in bold text. Quotations from the revised manuscript are shown in bold italics. Line numbers in the author responses refer to locations in the revised manuscript.

Referees' comments:

Referee #1 (Remarks to the Author):

Referees' comments:

Remarks to the Author:

The paper shows an interesting analysis of the effects of anthropogenic climate change on snowpack and, consequently, runoff in boreal regions. In general, I found the approach novel, providing new important results that are worth publishing in Nature. The assessment of human influence on snow cover evolution is carried out by using an innovative approach that is based on a training of an empirical model. Overall, the analysis is welcomed. The paper is also well written.

However, there are some issues affecting the reliability of presented results, thus requiring further consideration. A major issue is that the used observational data on snow water equivalent (SWE) is not really observational, but instead model predictions incorporating snow depth observations as model input. For example, the applied Canadian Meteorological Centre's Daily Snow Depth Analysis uses synoptic snow depth observations from weather stations as input to a model that predicts snow accumulation and melt (Brown & Brasnett, 2010). The more complex reanalysis models, also used here, do basically the same. But when snow depth observations are assimilated to models, problematic issues include how well is the snow density evolution considered when assessing the magnitude of SWE. As an outcome the ensemble of model predictions used here as 'observational' data set representing the peak SWE (1 April) may include systematic errors that seriously disturb the reliability of regional trends, see e.g. (Pulliainen et al. 2020) that discusses the reliability of SWE trends among different snow products. To consider this issue the authors should investigate and compare their data with the available SWE information including available historical distributed SWE observation data sets made in snow courses across Eurasia and Canada (see e.g., NSDC data archives). This comparison would evidently make the uncertainty analyses of the paper more convincing. Currently, the methods part of the paper indicates (L.346) that the SNOTEL network in the Western U.S. is used for the validation of the SWE reconstruction. This is not sufficient for the hemispheric analysis as the single point snow pillow measurements of SWE in the SNOTEL network are limited to mountainous regions of the Western U.S.

We thank the Reviewer for their thoughtful engagement with our work and appreciate their emphasis on how more incorporation of *in situ* data from a wider range of regions and topographies would strengthen the results we present. In response to this comment, we have made several substantive changes to the data informing our reconstruction and attribution, which improves the analysis and strengthens our original conclusions.

Briefly, we (1) include a wider array of *in situ* measurements, expanding the geographic scope and terrain diversity in our assessment; (2) we fold the *in situ* observations more fully

into a new hemispheric attribution analysis and the empirical reconstructions, allowing us to use them to detect and attribute forced SWE changes; and (3) we perform a more thorough validation of our empirical reconstructions vis-a-vis the *in situ* observations, including a comparison of long-term trends and evaluation of potential trends in biases.

(1) On the first, in addition to the data from the 550 SNOTEL sites used in the original manuscript, we include *in situ* SWE measurements from 341 locations in Canada from the Canadian historical Snow Water Equivalent (CanSWE) dataset (Vionnet et al., 2021), as well as 2,119 locations across the Northern Hemisphere the recently-published NH-SWE dataset. The latter makes use of more widely-available ground observations of snow depth (SD) and a SD-to-SWE conversion model that uses a regional parameterization based on climate variables to provide daily time series of SWE across the Northern Hemisphere (Fontrodona-Bach et al., 2023). While we recognize that NH-SWE, by converting depth to SWE, is a modeled product rather than a set of true instrumental measurements, the authors are able to estimate SWE with a high level of skill ($R^2 > 0.9$ for peak SWE for held-out evaluation data).

(2) On the second, inclusion of these additional datasets gives us near-hemispheric coverage (save Asia south of Russia, shown below in the revised Fig. 2a), with richer *in situ* measurements across a wider range of climates and topographies. The better data coverage now allows us to perform a new evaluation: in the revised manuscript we detect an anthropogenic fingerprint on the spatial pattern of observed *in situ* trends at the hemispheric scale using a climate model-based attribution technique (Padrón et al., 2020; Grant et al., 2021; Gudmundsson, Seneviratne, and Zhang, 2017; Qian and Zhang, 2015). Note that to perform this climate model-based attribution, we shift the quantity of interest from April 1 SWE to average March SWE due to both the greater availability of monthly SWE data from the climate models and the uneven sampling intervals of the expanded observations. The revised Figure 2 presents these attribution results, allowing us to confidently conclude that it is highly unlikely ($p < 0.01$) that the observed pattern of March SWE trends in these *in situ* measurements could have arisen from natural climate variability alone. This is consistent with the results we find applying the same attribution technique to gridded SWE products, and with our original results using the empirical reconstructions.

Figure 2. Trend in March SWE from 1981-2020 in in situ observations (a), the ensemble mean of 5 long-term gridded SWE products (b), and the multimodel mean of CMIP6 historical simulations with (c) and without (d) anthropogenic emissions. e, Spatial pattern correlation of 1981-2020 March SWE trends between the CMIP6 multimodel mean historical (red symbols) and historical-nat (blue symbols) simulations and each observational SWE product (see legend). The gray histogram indicates the empirical probability density function of spatial correlations between the historical trends and all possible 40-year trends from unforced pre-industrial control simulations ($N=78,601$). The red (orange) vertical dashed line indicates the 99th (95th) percentile of this empirical distribution.

(3) On the third analytical change, our expanded set of *in situ* measurements provides a more thorough validation of our empirical reconstructions. Using these additional data, we find that our models show consistent skill across the Northern Hemisphere, and in both mountainous and non-mountainous terrain (shown below in the revised Fig. S6).

Figure S6. R^2 (a) and RMSE (b) of Random Forest model predictions of *in situ* SWE at 2,961 locations over the period 1981-2020. Insets show the distribution of skill across sites, with the red line and value indicating the median.

Detailed line comments:

L.71-73: The possible change of bias with time in SWE products is an additional source of error that is not considered.

We thank the Reviewer for pointing out this additional source of potential error; our evaluation below suggests that it is not driving our results. To evaluate the degree to which time-varying bias influences our results, we calculate the annual bias of our empirical reconstructions with respect to the trustworthy *in situ* observations. Then, for each *in situ* location, we evaluate whether there are any time trends in annual bias using the same Theil-Sen slope estimator and Mann-Kendall test used for estimate trends in the rest of the analysis. We find that “*there are no systematic trends in time of the bias across our reconstructions relative to the *in situ* observations (shown below in revised Fig. S8), suggesting that the reconstruction models are capturing the real-world rate of change of snowpack with high fidelity*” (ll. 544-546).

Figure S8. Fraction of empirical reconstructions of *in situ* March SWE with statistically significant (Mann-Kendall $p < 0.05$) trends in bias relative to *in situ* observations.

L.83: References 25 and 26 only consider mountain regions (Western U.S. and European Alps).

Our expanded network of *in situ* observations includes both mountainous and non-mountainous terrain, and we consider additional work evaluating trends in Northern Eurasian snowpack (Bulygina et al., 2011) when interpreting our findings.

L.363: The SWE model used for reconstructing the historical April 1 SWE is simple, but robust and useful for the presented analysis.

We thank the Reviewer for appreciating the value of our reduced-form modeling approach.

L.368-370: The sentence ignores the limitations, as the paper does not currently employ the in situ SWE observations distributed across the Northern Hemisphere. References included here (25, 26, 59) only deal with regional analysis in some mountain regions (European Alps and Western U.S.) .

As noted above, our expanded network of *in situ* observations includes both mountainous and non-mountainous terrain, and we consider additional work evaluating trends in Northern Eurasian snowpack (Bulygina et al., 2011) when interpreting our findings. The original sentence has also been removed in the revised manuscript.

L.383: Results of Fig. S4 indeed capture the interannual variability of SWE quite well at the SNOTEL sites of the Western U.S. for the period 2000-2020 (i.e., 21 samples for each site). However, apparently any conclusion on trends cannot be made (or they are not shown)? Same kind of analysis could be made across the hemispheric scale using in situ snow course data (requires some interpolation as observations are typically available on a monthly, bi-monthly, or weekly basis for a given snow course).

We thank the Reviewer for the suggestion. Our updated analysis presented in the revised Figure 2 (presented above) makes use of hemispheric trends over the longer period 1981-2020 using *in situ* observations, and evaluates the consistency of those trends with anthropogenic forcing as derived from an ensemble of historical climate model simulations. Additionally, in response to this comment we evaluate the ability of our empirical models to reproduce the 1981-2020 trends at the *in situ* sites and find they can do so quite skillfully, with a pattern correlation of 0.72 between observed and reconstructed *in situ* trends (shown in the revised Fig. S7, reproduced below).

Figure S7. Observed in situ (a) and reconstructed (b) 1981-2020 March SWE trends at 2,961 locations. c, Scatterplot of reconstructed versus observed trends, where each dot represents an in situ location. Points are colored by their density. Dashed line denotes perfect agreement between reconstructed and observed trends. Pearson's correlation is shown in bottom right corner.

L.421: SWE-T-P evidently refers to temperature and precipitation, text would be less confusing (in general) if the number of abbreviations were reduced. Also e.g., abbreviation "NoACC" is somewhat problematic even though explained for the first time in L.99.

We thank the Reviewer for helping us clarify our communication. In the revised manuscript, we spell out temperature and precipitation, and no longer use the "NoACC" shorthand, instead referring to these reconstructions as "counterfactuals" and using the phrase "forced effect" instead of "ACC effect".

L.431: Uncertainty in SWE data products is not adequately considered concerning the possible temporally changing bias of these products, see the general comment above.

As noted in our response to detailed line comment 1 above, we now present an analysis of time-varying bias in our revised Fig. S8, where the vast majority of in situ locations have no time trends in the annual bias across our ensemble of reconstructions.

L.478: Since Eq. (6) is an implication of Eqs. (4) and (5) it could be expressed/clarified.

The compensatory precipitation analysis is no longer part of the revised manuscript.

L.498: It is unclear how the dSWE/dT curve is determined, as Eq. (7) is evidently providing a constant value for each basin?

In the revised analysis, we calculate the dSWE/dT curve separately for *in situ* observations, gridded observations, and climate models, in addition to our basin-scale reconstructions. We have also calculated it in a more straightforward manner simply by regressing March SWE on cold-season (November-March) average temperature across years at each location. We describe this in revised Methods section “Temperature sensitivity of snowpack”:

*“To better understand the drivers of the heterogeneous spatial response of SWE and its potential future changes with further warming, we evaluate the temperature sensitivity of March SWE across a gradient of climatological winter temperatures in *in situ* observations, gridded observations, our basin-scale reconstructions, and in climate models. The marginal effect of an additional degree of warming, $\frac{\Delta \text{SWE}}{\Delta T}$ or β , is calculated as the regression coefficient of March SWE on cold-season (November-March) temperature:*

$$\text{SWE}_{i,y} = \alpha_{i,0} + \alpha_{i,1} T_{i,y} \quad (3)$$

*Where $\text{SWE}_{i,y}$ is March SWE in unit i (*in situ* station, grid cell, or river basin) in water year y and $T_{i,y}$ is average winter temperature in that same unit. We run this regression at each *in situ* location, for all 20 combinations of gridded SWE and temperature products, for all 12 climate models (using the HIST simulations), and for all 120 basin-scale reconstructions. We then calculate the average and standard deviation of all of the coefficients for a given type of data (*in situ*, gridded observations, climate models, basin-scale reconstructions) in a rolling 5-degree temperature window to produce the curves in Figure 4a. As such, the uncertainty estimate includes both parametric and data uncertainty.” (ll. 629-643)*

Figures 1 and 2: Obtained numbers for the major basins/ivers discharging to the Arctic Ocean would be interesting to see as well (even though they represent sparsely populated areas).

We have calculated the average forced SWE change for these basins:

“[...] enhanced them in the cold, high-latitude basins that drain into the Arctic Ocean by $2.5 \pm 1.8\%$ per decade (Fig. 3e)” (ll. 180-181)

We also report the SWE-driven runoff changes for these basins:

“Snowpack in cold and sparsely-populated basins, meanwhile, is likely to be resilient to high levels of winter warming exceeding 5°C, such as that arising from Arctic amplification, and the coldest may see increased snowpacks and enhanced spring runoff into the Arctic Ocean of over 10% on average (Figs. 4b, S13h).” (ll. 266-269)

Figure S8: Reader can be confused with the multitude of abbreviations, such as AMJJ.

We thank the Reviewer for helping us clarify our communication. In the revised manuscript, we write out the months (e.g., April-July for AMJJ) and consistently refer to the period November-March as the “winter” and April-July as “spring”. Additionally, as mentioned above, we spell out temperature and precipitation, and no longer use the “NoACC” shorthand, instead referring to these reconstructions as “counterfactuals” and using the phrase “forced effect” instead of “ACC effect” (line # examples).

References:

Brown, R. & Brasnett, B. Canadian Meteorological Centre (CMC) Daily Snow Depth Analysis Data, Version 1. NASA National Snow and Ice Data Center Distributed Active Archive Center. (2010) doi:10.5067/W9FOYWH0EQZ3.

Pulliainen, J. et al. Patterns and trends of Northern Hemisphere snow mass from 1980 to 2018. *Nature*, 581:294-298 (2020).

Reviewed by Jouni Pulliainen

Response References:

Bulygina ON, Groisman PY, Razuvaev VN, et al. 2011. Changes in snow cover characteristics over Northern Eurasia since 1966. *Environmental Research Letters*. 6(4): pp.045204.

Fontrodona-Bach A, Schaepli B, Woods R, et al. 2023. NH-SWE: Northern Hemisphere Snow Water Equivalent dataset based on in situ snow depth time series. *Earth System Science Data*. 15(6): pp.2577–2599.

Gottlieb AR and Mankin JS. 2022. Observing, Measuring, and Assessing the Consequences of Snow Drought. *Bulletin of the American Meteorological Society*. 103(4): pp.E1041–E1060.

Grant L, Vanderkelen I, Gudmundsson L, et al. 2021. Attribution of global lake systems change to anthropogenic forcing. *Nature Geoscience*. 14(11): pp.849–854.

Gudmundsson L, Seneviratne SI and Zhang X. 2017. Anthropogenic climate change detected in European renewable freshwater resources. *Nature Climate Change*. 7(11): pp.813–816.

Padrón RS, Gudmundsson L, Decharme B, et al. 2020. Observed changes in dry-season water availability attributed to human-induced climate change. *Nature Geoscience*. 13(7): pp.477–481.

Qian C and Zhang X. 2015. Human Influences on Changes in the Temperature Seasonality in Mid- to High-Latitude Land Areas. *Journal of Climate*. 28(15): pp.5908–5921.

Vionnet V, Mortimer C, Brady M, et al. 2021. Canadian historical Snow Water Equivalent dataset (CanSWE, 1928–2020). *Earth System Science Data*. 13(9): pp.4603–4619.

Referee #2 (Remarks to the Author):

This study, by Gottlieb and Mankin, uses an ensemble of snow products and climate models to investigate climate change impacts on basin-scale snowpack and runoff in the northern hemisphere. The authors focus on April 1 SWE. They conclude that warming has reduced snow in the more highly populated basins and that future warming will have a worsening effect on SWE and runoff losses.

Overall, this study provides a nice dataset to investigate processes related to climate change. To me, it seems like an extension to the Mankin et al., 2015 paper (DOI 10.1088/1748-9326/10/11/114016) with similar conclusions. I like the attention to dataset uncertainties, but the manuscript tries to focus more on the runoff changes.

The methodology is OK, but the author's argue that snowpack assessments are quite sensitive to the data products analyzed and show disagreement between the products. However, they choose to use an ensemble approach with these products. My question is: why should we believe that the ensemble approach is better? Any mean value will be driven by the datasets that are further from the truth (especially with only a handful of datasets).

We thank the Reviewer for their thoughtful engagement with our work. We expand upon our motivation for the ensemble approach and why it is more defensible than using any one dataset alone.

Briefly, we only have observational truth about snowpack at a limited number of locations in the Northern Hemisphere, which are not necessarily representative of snow states or trends at the hydrologically-relevant scales (like river basins) at which water resource management decisions are made. Because of the imperative to assess snow changes under warming coupled with the limitations of *in situ* data to do so, we need to contend with other products and their differences. Assessing the gridded products against the *in situ* data can provide some guidance about relative product skill but only in places for which the *in situ* data exist. As such, we must return to the representativeness issue noted above: we want to make claims about snow changes at hydrologically meaningful scales, but do not have ground-based observations to do so. And so, we adopt an approach that is common to the climate modeling community, which is to recognize the observational products as an “ensemble of opportunity” rather than an ensemble of thoughtful design. In this framing, each product, like each model, has some truth and some noise in it. Our hope is to find the shared elements of truth among the observational products by treating each data combination as an equally-valid representation of truth. In other work in which we have taken this approach (Gottlieb and Mankin, 2022), we were able to show that leveraging all available SWE datasets to forecast historical warm-season droughts was over 100% more skillful than forecasts using any single product. These results suggest that the ensemble approach is indeed effective at isolating a signal amidst the inter-product noise. Changes to the text that reflect this argument more fully can be seen in the Methods section:

“Our ensemble approach is motivated by two main considerations. First, it is difficult to determine what represents “true” snowpack at hydrologically relevant scales. All methods of estimating spatially distributed snowpack (e.g., remote sensing or reanalysis) have their intrinsic limitations that result in high levels of disagreement on snow mass, its variability, and long-term trends (Mortimer et al., 2020; Gottlieb and Mankin, 2022), as we show in Figure 1. In situ measurements may represent truth at the locations at which they are collected, but are difficult to generalize, especially in complex terrain. As a result, using these point observations to adjudicate which gridded products (whose values represent averages over tens to tens of thousands of kilometers) lie closest to “truth” is challenging. Given the inability to know the true state of snowpack or rigorously rule out any of its various gridded estimates, we choose to consider these observational products as equally valid estimates of truth in which we can attempt to identify shared responses. Second, the ensemble approach allows us to capture the structural uncertainty in how SWE responds to changes in temperature and precipitation, which are themselves subject to data uncertainties (Fig. S9). Using all dataset combinations, we can sample and characterize uncertainty in SWE, temperature, and

precipitation and their covariance with one another. Such an approach has been used to estimate forced changes in components of the Earth system in which both the dependent and independent variables of interest are themselves uncertain (Yao et al., 2023).” (ll. 513-529)

Furthermore, the largest RMSE values compared to SNOTEL (50% or greater, Fig. S4) end up being in the southwestern US in which much of the discussion is focused. Thus, I am not convinced that this ensemble is appropriately representing the analysis that the authors are attempting.

In response to this and other concerns about the skill of our ensemble reconstructions, we adopt a revised empirical approach to generating the reconstructions on which our conclusions are based. Our revised approach is far more skillful than that presented in our original submission and greatly strengthens many of the original conclusions of the work.

Briefly, our revised method now uses a machine learning (Random Forest) model in place of the multiple regression, allowing for greater model flexibility. The Random Forest algorithm predicts March SWE using monthly-scale temperature and precipitation from November through March, versus seasonal averages in the original, and is trained on the full spatiotemporal panel of data (i.e., all grid cell-years), versus individually at each grid cell in the original. This results in greatly improved model skill relative to the original multivariate OLS model, both in reproducing the gridded SWE products (revised Fig. S3, S5) and in predicting out-of-sample *in situ* SWE (revised Fig. S6, S7), including in the Southwestern U.S. where we identify large changes. We reproduce those figures below to show the skill of the reconstructions in the revised manuscript.

Figure S3. Basin-scale cross-validated R^2 (a-e) and root-mean-square error (RMSE; f-j) for 5 gridded SWE data products over the period 1981-2020. Each metric shows the skill of the mean of all reconstructions for a single SWE product versus the observed values from that product. Insets show the distribution of skill across basins, with the red line and value indicating the median. See Methods for cross-validation procedure.

Figure S5. Observed (a-f) and reconstructed (g-l) 1981-2020 March SWE trends for 5 gridded SWE data products and their mean. m-r, Scatterplot of reconstructed versus observed trends, where each dot represents a river basin. Dashed line denotes perfect reconstruction. Pearson's correlation is shown in bottom right corner.

Figure S6. R^2 (a) and RMSE (b) of Random Forest model predictions of in situ SWE at 2,961 locations over the period 1981-2020. Insets show the distribution of skill across sites, with the red line and value indicating the median.

Figure S7. Observed in situ (a) and reconstructed (b) 1981-2020 March SWE trends at 2,961 locations . c, Scatterplot of reconstructed versus observed trends, where each dot represents an in situ location. Points are colored by their density. Dashed line denotes perfect agreement between reconstructed and observed trends. Pearson's correlation is shown in bottom right corner.

Additionally, the authors reconstruct snowpack as a function of temperature and precip., with temperature driving much of the changes. However, a recent paper has shown that temperature thresholds alone give large errors in precipitation phase estimates for the northern hemisphere, thus adding uncertainty to some of the main points about precipitation phase near a melting point threshold.

Reference: Jennings et al., 2018: <https://www.nature.com/articles/s41467-018-03629-7>

We now clarify in our Methods that we do not impose any temperature thresholds for rain-snow partitioning or snowmelt in our empirical modeling:

"Additionally, this empirical approach imposes no a priori assumptions about temperature thresholds for rain-snow partitioning or snowmelt, which can vary substantially in space and are themselves a contributor to uncertainty in modeled estimates of SWE (Kim et al., 2021; Jennings et al., 2018)." (ll. 497-499)

Our discussion of thresholds in the original was intended to provide some physical and statistical intuition for the emergent nonlinearity in the temperature sensitivity of snow (revised Fig. 4a, original Fig. 3). We have edited this section to make the physical drivers of the generalizable nonlinear sensitivity of SWE to temperature more clear:

"There are several notable features of these curves. First, is their scale- and data-invariance: the location of the inflection point in temperature sensitivity is consistent when it is estimated from point measurements, gridded data products, climate models, or our basin scale reconstructions. This consistency suggests that despite substantial measurement and modeling uncertainties, simple thermodynamics can explain much of snow's historical and future response to warming. As a location's climatological temperature warms towards the freezing point, the likelihood of subseasonal temperatures exceeding thresholds where precipitation is partitioned towards rain over snow or accumulated snowpack will melt increases exponentially. We note, however, that these thresholds themselves are not constant in space, owing to factors such as topography and distance from oceanic moisture sources (Jennings et al., 2018), which may account for some of the uncertainty in snow sensitivities at any one climatological temperature." (ll. 220-229)

I would like to emphasize that I like the general approach in this study and that these comments are meant to be constructive. However, some improvements based on my comments above/below are required for publication, in my opinion.

Other minor comments:

the term "boreal": the term boreal is used in the title and elsewhere, but I disagree with its use. In the snow hydrology community, boreal areas are generally between 50 and 60 or 65 degrees latitude whereas the authors appear to use it as everything south of 60 degrees.

We no longer use the term "boreal" to refer to the entire Northern Hemisphere in either the title, which has been changed to "Attributing Northern Hemisphere snow loss and its consequences to human influence" or the text.

line 31-33: if the data products are sensitive to the scale and analysis performed, what datasets may not be appropriate for the scale and analysis that was performed here?

This sentence has been removed from the revised manuscript.

lines 40-42: These are also the basins with some of the highest RMSE.

Our revised Random Forest classifier reduces inter-basin heterogeneity in skill that prompted this comment—our reconstructions are now far more skillful in these populous basins (Figs. S3, S4, S6).

line 65: this statement "snowpacks across the Northern Hemisphere" is different than what the title implies.

This statement now reads:

“Together, our results provide a thorough documentation of the historical and future effects of climate change on snow water storage at the hemispheric and river basin scales.” (ll. 64-65)

Additionally, we now make the scale of the analysis being discussed throughout the manuscript more clear.

Line 70: Not sure that I am buying it with that many large RMSE values.

The Random Forest model used in the revision reduces inter-basin heterogeneity in skill that prompted this comment—our reconstructions are now far more skillful (Figs. S3, S4, S6).

line 151 (and elsewhere): please refrain from using terms like "very" as these are not well defined metrics (i.e., how cold is very cold?).

We have removed these and other ambiguous adverbs.

line 199-200: this sentence is a bit distracting for the conclusions section, in my opinion.

We have removed this sentence.

line 208-209: Please further expand on the quantifications of uncertainty sources. I gained little to nothing from the text about this and only a bit when looking at the figure, but nothing well quantified.

In response to this comment, we have both simplified the uncertainty quantification by focusing on fractional uncertainty (e.g. the relative proportion attributable to each source; Hawkins and Sutton, 2009; Lehner et al., 2020) to make it more easily interpretable, and integrated it into the text where appropriate.

The Methods section “Uncertainty partitioning of forced SWE changes” details the fractional uncertainty quantification:

“To quantify the magnitude of uncertainty introduced by each source, we calculate the standard deviation of forced SWE trends across a single dimension, holding all others at their mean. For instance, the uncertainty due to differences in model structure is given by the standard deviation of forced SWE trends across the 12 climate models (considering only the first realization from each), taking the mean across all SWE-temperature-precipitation dataset combinations.

To isolate the uncertainty from internal variability in temperature and precipitation, we use 50 pairs of historical and historical-nat simulations from the MIROC6 model, which differ only in their initial conditions. We take the standard deviation of forced SWE trends for all 50 realizations of forced changes in temperature and precipitation separately, taking the mean across all SWE, temperature, and precipitation data product combinations.

Consistent with previous work in uncertainty partitioning (Hawkins and Sutton, 2009, 2011; Lehner et al., 2020), we consider total uncertainty T in the forced SWE trend in basin b to be the sum of all four sources:

$$\sigma_b = \sigma_S + \sigma_{TP} + \sigma_M + \sigma_I \quad (3)$$

where S is the uncertainty from SWE observations, TP is the uncertainty from temperature and precipitation observations, M is the uncertainty from model structure, and I is the uncertainty from internal variability. To assess which sources are the largest contributor to uncertainty in each basin, we consider the fractional uncertainty of each (e.g. $\frac{\sigma_S}{\sigma_b}$ gives the proportion of uncertainty in basin b attribution to SWE observational uncertainty). This fractional uncertainty is reported in Figure S12. For each source, we hatch out basins where the magnitude of uncertainty is insufficient to change the sign of the ensemble mean estimate of the forced SWE trend (i.e., the signal-to-noise ratio is >1).” (ll. 608-626)

The presentation of the fractional uncertainty partitioning is presented in the revised Figure S12, reproduced below. We now more concretely draw on this analysis in our discussion of the value of uncertainty partitioning:

“Additionally, there is value in identifying and quantifying these sources of uncertainty in forced snowpack changes (Fig. S12), as it can guide future scientific and operational decision-making. For instance, uncertainty in the forced response of temperature and precipitation arising from structural differences between climate models is the dominant source of uncertainty in the magnitude of forced March SWE trends in over half (95 out of 169) of all basins (Fig. S12a, d), suggesting that improving the skill of climate models in capturing regional climate would go a long way towards constraining historical and future snow change. Uncertainty in SWE data products themselves is also a limiting factor in many basins where in situ observations are sparse or non-existent (Fig. S12a, b), suggesting that constraining observational estimates of SWE would be most valuable. Finally, identifying the contribution of irreducible uncertainty in SWE trends from internal variability in the climate system (Fig. S12e) also has considerable value, as it indicates the range of physically consistent snowpack trajectories for which water resource managers and stakeholders must be prepared.” (ll. 299-310)

Additionally, we draw on these insights in our discussion of the influence of internal variability on our results:

“Interestingly, we are able to detect a forced SWE decline in major basins such as the Columbia (4.8% per decade) where historical observations suggest modest increases since 1981 or the Saint Lawrence (6.9% per decade), where observed trends have been small and statistically insignificant. Together these examples suggest that internal variability in the climate system has been masking large forced snowpack reductions in some regions. Likewise, there are basins like the Rio Grande, which have suffered large historical snowpack declines of over 10% per decade, but for which there is little agreement that forced temperature and precipitation changes have caused those declines, reinforcing the notion that low-frequency variability can overwhelm forced signals in snow and hydroclimate, even on multidecadal timescales. Indeed, internal variability is the dominant source of uncertainty in the forced response—over climate model structural differences and observational uncertainty in SWE, temperature, and precipitation—in roughly 1 in 8 basins (Fig. S12).” (ll. 182-192)

line 216-225: this is all discussion and nothing that resembles a conclusion. Please move to an appropriate section of the article.

We have removed these points from the Conclusion, and have changed the title of this final section from “Conclusions” to “Managing and leveraging uncertainty” to make it clear that we are not simply summarizing our results, but synthesizing the lessons learned in the research.

Figure S12. *a*, Dominant source of uncertainty in reconstruction-based estimates of forced March SWE trends from 1981 to 2020. *b-e*, Percentage of total uncertainty in forced SWE trends attributable to (*b*) observational uncertainty in gridded SWE products, (*c*) observational uncertainty in temperature and precipitation data products, (*d*) uncertainty in the forced response of temperature and precipitation across different climate models, and (*e*) uncertainty in the forced response of temperature and precipitation arising from internal variability (Methods). Hatching indicates basins where the uncertainty attributable to a given source is insufficient to change the sign of the ensemble mean estimate of the forced SWE trend.

Figures: Please change the projection. The current projection focuses on mid-latitudes and "squeezes" much of the northernmost basins.

All figures now use a projection in which the high-latitude basins are more easily discerned.

Response References:

Gottlieb AR and Mankin JS. 2022. Observing, Measuring, and Assessing the Consequences of Snow Drought. *Bulletin of the American Meteorological Society*. 103(4): pp.E1041–E1060.

Hawkins E and Sutton R. 2011. The potential to narrow uncertainty in projections of regional precipitation change. *Climate Dynamics*. 37(1): pp.407–418.

Hawkins E and Sutton R. 2009. The Potential to Narrow Uncertainty in Regional Climate Predictions. *Bulletin of the American Meteorological Society*. 90(8): pp.1095–1108.

Jennings KS, Winchell TS, Livneh B, et al. 2018. Spatial variation of the rain–snow temperature threshold across the Northern Hemisphere. *Nature Communications*. 9(1): pp.1148.

Kim RS, Kumar S, Vuyovich C, et al. 2021. Snow Ensemble Uncertainty Project (SEUP): quantification of snow water equivalent uncertainty across North America via ensemble land surface modeling. *The Cryosphere*. 15(2): pp.771–791.

Lehner F, Deser C, Maher N, et al. 2020. Partitioning climate projection uncertainty with multiple large ensembles and CMIP5/6. *Earth System Dynamics*. 11(2): pp.491–508.

Mortimer C, Mudryk L, Derksen C, et al. 2020. Evaluation of long-term Northern Hemisphere snow water equivalent products. *The Cryosphere*. 14(5): pp.1579–1594.

Yao F, Livneh B, Rajagopalan B, et al. 2023. Satellites reveal widespread decline in global lake water storage. *Science*. 380(6646): pp.743–749.

Referee #3 (Remarks to the Author):

This study evaluates historical trends in basin-scale snowpack (SWE) and consequent snow-driven spring runoff over Northern Hemisphere during 1981-2020 using a statistical reconstruction approach. The authors argue that they have identified human-caused changes in SWE and consequent runoff (“attribution” in their definition) by comparing 40-year trends in the observation-based SWE reconstructions (“historical”) with those in the counterfactual reconstructions without human influence (“NoACC”).

I have focused on evaluating the attribution method as requested by the Editor. The attribution approach used in this paper has three steps:

First, the authors construct a multiple linear regression model (Eq. 1) of reconstructing (or predicting) April 1 SWE at each basin using cold season (NDJFM) temperature (T) and precipitation (P). They train this regression model using observational datasets during 2001-2020, where 8 SWE, 4 temperature (T), and 5 precipitation (P) datasets (including reanalyses and satellite measurements) are all combined to consider observational uncertainties, providing 160-member ensemble of observations and corresponding regression equations.

Second, for each observational-based member, counterfactual SWE reconstructions are estimated by inserting adjusted T and P into the pre-defined regression model. Here, T and P values are adjusted by removing possible anthropogenic changes from the observations. The anthropogenic changes in T and P are obtained from differences between historical (ALL) and historicalnat (NAT) simulations of available CMIP6 multi-model simulations (86 ALL-NAT values available).

Finally, the authors compare a distribution of 'historical' SWE trends (consisting of 160 observation-based estimates) with that of 'NoACC' SWE trends (consisting of 13,760 estimates obtained from 160 corresponding estimates \times 86 CMIP6 ALL/NAT runs). To claim that "historical SWE trends are attributable to human causes", the authors assess the significance of difference of the two SWE trend distributions using a Kolmogorov-Smirnov test (5% level) and also check the sign agreement of trend difference (ALL-NAT > 0 in more than 75% of 86 runs) if my understanding is correct.

Applying this procedure to all snow-dependent river basins, the authors conclude that they can "confidently attribute changes in April 1 SWE to human causes in roughly 40 percent of the world's major river basins, including the 4 most populous snow-dominated basins in Europe and 3 of the top 4 in North America". They employ a similar statistical approach (Eq. 4) to further estimate snowpack-driven changes in spring runoff in the past and future conditions.

I find this study interesting in that it comprehensively takes account of observational uncertainties through SWE-T-P data combinations at the river basin scales (step 1 above). However, with regard to its attribution analysis (steps 2 and 3 above), I do not think that this approach can provide a way to claim "robust attribution of human-forced changes". Conventionally, attribution approaches quantify external forcing contributions to the 'observed' changes by either checking if the model-simulated forced pattern is present in the 'observations' (typically based on regressions but also using similarity measures) or

comparing the likelihoods of the 'observed' changes between real-world and counterfactual conditions (similar to event attribution types of assessments). In contrast to the usual definition of attribution, this study evaluates human influences by comparing two estimates of reconstructed SWE trends, differences of which are largely determined by CMIP6-simulated T and P changes (ALL-NAT), irrespective of the significance and sign of 'observed' trends. In particular, cold season mean T differences between ALL and NAT (using their 30-year means) will induce SWE differences almost linearly based on the statistical relation in Eq. (1). In this way, resulting SWE differences likely reflect model response patterns, not indicating that this pattern is present in the observations or contributes to the observed changes. Since no observational information is used in this trend comparisons, my interpretation of their 'attribution' results would be such that CMIP6-simulated T and P responses to anthropogenic forcing are larger than observational uncertainties in T and P over many river basins or similar.

We thank the Reviewer for their thorough and thoughtful synthesis and constructive comments about our attribution approach. In response to this and the above Reviewer comments, we make several important analytical changes.

First, on the attribution approach: In the revised manuscript, we now perform a substantial new attribution analysis at the hemispheric scale, in which we compare model-simulated SWE trends to observed trends. We use a widely-used model attribution approach (Qian and Zhang, 2015; Gudmundsson, Seneviratne, and Zhang, 2017; Padrón et al., 2020; Grant et al., 2021), asking the degree to which the observed pattern of March snowpack trends is consistent with the pattern from anthropogenic forcing as simulated by the models, while taking into account the uncertainty in the observations themselves. We describe this approach in detail in the new Methods section "Attributing SWE trends to anthropogenic forcing", which we reproduce here:

"Our hemispheric attribution approach tests whether the similarity between observed and climate model-simulated forced SWE trends exceeds what could be possible from natural climate variability alone. To evaluate the null hypothesis that the pattern of SWE trends in the HIST simulations could be the result of natural variability alone, we calculate the trend in March SWE from 1981 to 2020 in each model's HIST simulation and for every unique 40-year period from those same models' unforced PIC simulations (e.g., for a 500-year PIC simulation, we generate 461 maps of 40-year trends). All trends are calculated using the Theil-Sen estimator, a non-parametric technique for estimating a linear trend that is more robust to data that is skewed or contains outliers than ordinary least squares (OLS). Then, we calculate the Spearman (rank) correlation coefficient between the maps of HIST and PIC trends to quantify the pattern similarity. The resulting empirical distribution of 78,601 correlations (background histogram on Figure 2) represents the likelihood that the pattern in the forced historical simulations could have arisen from natural variability alone.

We quantify the similarity between observed SWE trends and the model-estimated response to forcing by taking the Spearman correlation between each observational product (Table S1) and the multimodel mean of the HIST simulations (red symbols in Figure 2e). For this analysis, the in situ observations are aggregated to the same 2°x2° grid as the gridded observations and climate models by taking the mean trend of all stations within each grid cell (Figure 2a). If the correlations between the observations and HIST simulations are greater than almost all of the correlations between the HIST and PIC simulations, we can reject the null hypothesis that the observed historical pattern could have arisen from natural variability alone and claim that a response to historical forcing is present in the observed pattern. Furthermore, if we cannot reject the null hypothesis using the correlations between the observations and HIST-NAT simulations with only solar and volcanic forcing, then it is unlikely that the observed pattern is the result of natural radiative forcing. Combined, these two lines of evidence would strongly suggest that anthropogenic forcing is causing the observed pattern of SWE trends.” (ll. 463-487)

The results are presented in the revised Figure 2, which is reproduced below. These results make clear that anthropogenic emissions have contributed to the observed pattern of March SWE trends from 1981 to 2020, as it is exceedingly unlikely that such a pattern could have arisen from natural variability alone. Crucially, we show that this finding holds for both a dramatically expanded set of *in situ* observations and our long-term gridded SWE products, with the exception of the MERRA-2 reanalysis.

Figure 2. Trend in March SWE from 1981-2020 in in situ observations (a), the ensemble mean of 5 long-term gridded SWE products (b), and the multimodel mean of CMIP6 historical simulations with (c) and without (d) anthropogenic emissions. e, Spatial pattern correlation of 1981-2020 March SWE trends between the CMIP6 multimodel mean historical (red symbols) and historical-nat (blue symbols) simulations and each observational SWE product (see legend). The gray histogram indicates the empirical probability density function of spatial correlations between the historical trends and all possible 40-year trends from unforced pre-industrial control simulations ($N=78,601$). The red (orange) vertical dashed line indicates the 99th (95th) percentile of this empirical distribution.

Second, we pursue an attribution at the river basin scale, as in the original manuscript, in which we employ a data-model fusion approach in which we empirically model SWE as a function of temperature and precipitation, then use climate models to estimate how anthropogenic forcing has affected regional temperature and precipitation, and by extension SWE. We feel it is necessary to move beyond the more traditional model-based attribution approach presented in the revised Fig. 2 to assess how anthropogenic climate change has affected basin-scale snow trends, as the climate models are limited in their ability to capture the detail of SWE change at these finer scales. We articulate this motivation more clearly as the text transitions to focus on the reconstruction-based analysis:

“While coupled climate model experiments such as those presented in Figure 2 are a powerful tool for detecting and attributing human influence on the broad features of the hemispheric pattern of SWE trends, the ability of these models to capture the magnitude and detailed spatial structure of observed trends is limited (see the range of the x-axis in Figure 2e), undermining the ability to assess forced snow change and its consequences at impacts-relevant scales.” (ll. 116-120)

In addition to an expanded list of citations of papers that have used a similar “observations minus modeled forced response of independent variables” approach to attribution (e.g., (Abatzoglou and Williams, 2016; Williams et al., 2020; Williams, Cook, and Smerdon, 2022; Yao et al., 2023; Diffenbaugh, Davenport, and Burke, 2021; Callahan and Mankin, 2022)), we have made key changes and perform additional analyses that strengthen our confidence in the data-model fusion approach. In our original submission, the empirical model was a multivariate regression. As noted above, we now use a Random Forest model using monthly temperatures and precipitation from November through March to more flexibly and skillfully predict March SWE. Additionally, whereas the multivariate regression models were fit individually for each grid cell, we train the Random Forest model on the full spatiotemporal panel of data (i.e., all grid cell-years from 1981-2020). This significant expansion of the support of the data on which the model is trained results in much-improved model skill (see revised Figs. S3-S5), which reduces the uncertainty in the attribution that arises from deficiencies in the skill of the empirical model. As we discuss in greater detail shortly in response to the Reviewer’s insightful comments about the stability of the empirical models, the expanded support of the data seems to result in estimates that are quite statistically stable (revised Fig. S4).

Finally, we take the additional step of showing that the empirical approach can produce reliable estimates of a forced SWE response by using the climate model simulations (in which we know the “true” counterfactual, unlike the real world):

“For each model, we fit the model described in Equation 1 using SWE, temperature, and precipitation data from the HIST simulations over the 1981-2020 period. Then, we use the empirical model trained on HIST data to predict counterfactual SWE using temperature and precipitation from the HIST-NAT simulations. Finally, we compare the forced (HIST minus HIST-NAT) trends calculated from the reconstruction approach to the “true” forced trends calculated using climate model SWE from the HIST and HIST-NAT experiments (Fig. S13). The strong similarity in the patterns of the “true” and reconstructed forced responses suggests that using observations with forced changes in temperature and precipitation removed produces reasonable estimates of a forced SWE change.” (ll. 590-598)

Both of these attribution changes (hemispheric and basin scale) are possible due to the expanded in situ data we use, and our change to using a very skillful Random Forest algorithm trained on all data rather than locally-fit linear regression for our reconstructions.

Figure S3. Basin-scale cross-validated R^2 (a-e) and root-mean-square error (RMSE; f-j) for 5 gridded SWE data products over the period 1981-2020. Each metric shows the skill of the mean of all reconstructions for a single SWE product versus the observed values from that product. Insets show the distribution of skill across basins, with the red line and value indicating the median. See Methods for cross-validation procedure.

Figure S4. RMSE in the 10 coldest, 10 warmest, and 20 “average” years in each basin from 1981-2020. Boxplots show the distribution of skill across basins in each temperature category, with the line indicating the median basin skill, the box the interquartile range, and the whiskers the 2.5th and 97.5th percentiles.

Figure S5. Observed (a-f) and reconstructed (g-l) 1981-2020 March SWE trends for 5 gridded SWE data products and their mean. M-r, Scatterplot of reconstructed versus observed trends, where each dot represents a river basin. Dashed line denotes perfect reconstruction. Pearson's correlation is shown in bottom right corner.

Figure S14. Forced (HIST minus HIST-NAT) trends in March SWE from 1981-2020 based on (a) climate model SWE output and (b) counterfactual SWE estimated using HIST-NAT temperature and precipitation and Random Forest model. C, Scatterplot of reconstructed versus original trends, where each dot represents a grid cell. Points are colored by their density. Dashed line denotes perfect agreement between reconstructed and original trends. Pearson's correlation is shown in upper left corner.

I have another concern about the stability of their statistical models. The statistical reconstruction method assumes that SWE dependence on T and P will not change with time but the validity of this assumption is not discussed. Statistical models are trained using recent SWE, T, and P datasets for 2001-2020, from which regression coefficients (betas) are estimated. In this setting, SWE variations will be mainly associated with inter-annual variabilities in T and P during the short period. The authors, however, use the same betas to estimate observed SWE reconstructions for the early period for 1981-2000 as well as to estimate SWEs for much colder pre-industrial (for NoACC) and much warmer future conditions (+1 degree and even near the late 21st century). This extrapolation of statistical relation may induce large errors in SWE estimates. Same issue applies to spring runoff reconstructions in Eq. (4).

The Reviewer is right to note that the relationship between snow and its drivers may not be time invariant. In response to this comment, we have made several important analytical changes to better accommodate the varying relationship between snow and its drivers in space and time. First, to avoid needing to extrapolate back in time, we use only datasets with complete coverage over the 1981-2020 period (ERA5-Land, JRA-55, MERRA-2, TerraClimate, Snow-CCI, and *in situ* data). Additionally, in the empirical modeling, we no longer fit a multiple regression model at each grid cell; instead, we use a Random Forest machine learning algorithm trained on the full spatiotemporal panel of data (i.e., all grid cell-years from 1981 to 2020). This significant expansion of the support of the data on which the model is trained results not only in much-improved model skill (Fig. S3-S5), but estimates that appear to be quite stable. To quantify this, we look at the RMSE in the hottest and coldest 10 winters in each basin, as well as the 20 “average” years in between, and find no differences in skill across this temperature gradient (revised Fig. S4).

Figure S3. Basin-scale cross-validated R^2 (a-e) and root-mean-square error (RMSE; f-j) for 5 gridded SWE data products over the period 1981-2020. Each metric shows the skill of the mean of all reconstructions for a single SWE product versus the observed values from that product. Insets show the distribution of skill across basins, with the red line and value indicating the median. See Methods for cross-validation procedure.

Figure S4. RMSE in the 10 coldest, 10 warmest, and 20 “average” years in each basin from 1981-2020. Boxplots show the distribution of skill across basins in each temperature category, with the line indicating the median basin skill, the box the interquartile range, and the whiskers the 2.5th and 97.5th percentiles.

Figure S5. Observed (a-f) and reconstructed (g-l) 1981-2020 March SWE trends for 5 gridded SWE data products and their mean. M-r, Scatterplot of reconstructed versus observed trends, where each dot represents a river basin. Dashed line denotes perfect reconstruction. Pearson's correlation is shown in bottom right corner.

The other issue related to attribution is about the datasets used. Since many observational datasets are based on estimates from reanalyses and satellite measurements rather than real observations, it looks difficult to attribute historical changes.

As noted in response to comments from Reviewers 1 and 2, we employ a much larger set of *in situ* observations in our hemispheric attribution analysis, as well as in the empirical reconstructions.

In addition to the SNOTEL data used in the original manuscript, we include *in situ* SWE measurements from the Canadian historical Snow Water Equivalent (CanSWE) dataset (Vionnet et al., 2021), as well as the recently-published NH-SWE dataset. The latter makes use of more widely-available ground observations of snow depth (SD) and an SD-to-SWE conversion model that uses a regional parameterization based on climate variables to provide daily time series of SWE across the Northern Hemisphere (Fontrodona-Bach et al., 2023). Combined, these data give us near-hemispheric coverage (save Asia south of Russia) of about 3,000 *in situ* measurements, which allows us to evaluate whether we can detect an anthropogenic fingerprint on the spatial pattern of those observed trends. The consistency of the attribution analysis using these *in situ* observations and the gridded products is strong evidence that we have detected and attributed a shared anthropogenic signal.

Regarding climate model simulations, the authors do not evaluate model performances at all. Depending on locations and seasons, models can systematically overestimate or underestimate T and P climatology and their responses to anthropogenic forcing.

For our estimate of counterfactual temperature and precipitation, we use a “delta” method where we calculate the difference between the HIST and HIST-NAT simulations and remove that from the observations; as such, biases in climatology should not matter, as each model is benchmarked to its own climatology. We now include an expanded list of references that have used a similar delta approach to estimate a forced response to be removed from observations in an attribution context (Abatzoglou and Williams, 2016; Williams et al., 2020; Williams, Cook, and Smerdon, 2022; Yao et al., 2023; Diffenbaugh, Davenport, and Burke, 2021; Callahan and Mankin, 2022) and have clarified the approach in the Methods text:

“We calculate the temperature response to anthropogenic forcing as the difference between the 30-year rolling mean average temperature for each month in the HIST and HIST-NAT runs. For precipitation, we calculate the forced response as the percentage difference between 30-year rolling mean monthly precipitation in HIST versus HIST-NAT. By differencing experiments from the same model, we hope to limit the influence of model biases in climatological temperature and precipitation, as each model is benchmarked to its own climatology [. . .] Having estimated anthropogenically-forced changes in gridded temperature and precipitation, we create counterfactual time series of temperature and precipitation by downscaling the output to the 0.5°x0.5° resolution of the observational ensemble using conservative regridding and removing the forced response from each model realization from

each gridded temperature and precipitation dataset. Temperature is adjusted by subtracting the forced change from the observations and precipitation is adjusted by the forced percentage change.” (ll. 556-575)

Systematic biases in the model-simulated trends (e.g., too rapid warming or wetting) could be a problem not addressed by the delta method we use. For example, trend biases could lead us to over- or under-estimate the forced response. To address this possibility, we now evaluate model biases in the 1981-2020 trends in winter temperature and precipitation (revised Fig. S9), and report the results in the main text when introducing the counterfactual SWE reconstruction:

“We note that the CMIP6 models tend over-estimate the historical warming trend compared to observations in some regions, particularly over Central North America and Eastern Europe (Fig. S9c, S10). At the same time, however, fewer than 1% of apparent biases over the hemisphere fall outside the range of model internal variability, suggesting that models are skillfully capturing North Hemisphere wintertime land temperature trends³⁸We note that the CMIP6 models tend to simulate slightly too strong of a historical warming trend compared to observations, most notably over Central North America and Eastern Europe (Fig. S9c, S10), though fewer than 1% of apparent biases over the hemisphere fall outside the range of model internal variability, which is an important consideration in model evaluation (Jain et al., 2023). The models also underestimate the drying in the Southwestern U.S., which has seen historical precipitation declines driven by both internal ocean-atmosphere variability and anthropogenic forcing (Williams et al., 2020), and underestimate observed wetting over the Tibetan Plateau (Fig. S9f, S10), though once again, fewer than 3% of precipitation biases are inconsistent with model internal variability (Jain et al., 2023). The models also underestimate the drying in the Southwestern U.S., which has seen historical precipitation declines driven by both internal ocean-atmosphere variability and anthropogenic forcing¹¹, and underestimate observed wetting over the Tibetan Plateau (Fig. S9f, S10), though once again, fewer than 3% of precipitation biases lie outside that possible from modeled internal variability, suggesting these biases do not undermine our attribution.” (ll. 167-175)

Additionally, we have added text to the Methods detailing how we evaluate model biases and their influence on SWE changes:

“Systematic biases in the model-simulated trends (e.g., too rapid warming or wetting), however, could potentially lead to over- or under-estimate the forced response. To address this possibility, we evaluate model biases in the 1981-2020 trends in winter temperature and precipitation against observed trends by taking the difference between the CMIP6 HIST ensemble mean and the mean of the observational products for each quantity (Fig. S9). To test whether the observed and modeled trends are consistent, we ask whether the observed trend

falls within a plausible range of forcing plus internal variability, given as the 2.5-97.5th percentile of the CMIP6 HIST trends. Only 1% (3%) of grid cells fall outside this range for temperature (precipitation), suggesting that the climate models capture realistic historical climate trends at these scales.” (ll. 561-569)

Figure S9. Observed trends in November-March average temperature (a) and total precipitation (d) from 1981 to 2020. B, e. Ensemble mean of historical CMIP6 simulations. C, f. Average bias in trends across all observation-model combinations. Hatching indicates regions where the observed trend falls outside the 2.5-97.5th percentile range of the CMIP6 trends.

In this respect, Table S2 indicates that a single model provides 50 members out of total 86 members, potentially influencing results depending on the model’s biases.

In response to this comment, we repeated the analysis using only the first realization from each climate model and found virtually identical results (revised Fig. S12), suggesting a limited influence of realization numbers in disproportionately weighting a single model in the overall response.

Figure S13. As in Figure 3, but using only the first ensemble member from each climate model to estimate counterfactual.

Overall, while detailed and useful information on snowpack and associated runoff changes due to warming is provided at river basin scales, I do not agree that this study provides robust attribution results due to the limitation of the statistical reconstruction approach.

We thank the Reviewer for their comments as they have greatly strengthened the analysis and the clarity and robustness of our original claims.

Response References:

Abatzoglou JT and Williams AP. 2016. Impact of anthropogenic climate change on wildfire across western US forests. *Proceedings of the National Academy of Sciences*. 113(42): pp.11770–11775.

Callahan CW and Mankin JS. 2022. National attribution of historical climate damages. *Climatic Change*. 172(3): pp.40.

Diffenbaugh NS, Davenport FV and Burke M. 2021. Historical warming has increased U.S. crop insurance losses. *Environmental Research Letters*. 16(8): pp.084025.

Fontrudona-Bach A, Schaeffli B, Woods R, et al. 2023. NH-SWE: Northern Hemisphere Snow Water Equivalent dataset based on in situ snow depth time series. *Earth System Science Data*. 15(6): pp.2577–2599.

Grant L, Vanderkelen I, Gudmundsson L, et al. 2021. Attribution of global lake systems change to anthropogenic forcing. *Nature Geoscience*. 14(11): pp.849–854.

Gudmundsson L, Seneviratne SI and Zhang X. 2017. Anthropogenic climate change detected in European renewable freshwater resources. *Nature Climate Change*. 7(11): pp.813–816.

Hawkins E and Sutton R. 2011. The potential to narrow uncertainty in projections of regional precipitation change. *Climate Dynamics*. 37(1): pp.407–418.

Hawkins E and Sutton R. 2009. The Potential to Narrow Uncertainty in Regional Climate Predictions. *Bulletin of the American Meteorological Society*. 90(8): pp.1095–1108.

Jain S, Scaife AA, Shepherd TG, et al. 2023. Importance of internal variability for climate model assessment. *npj Climate and Atmospheric Science*. 6(1): pp.1–7.

Lehner F, Deser C, Maher N, et al. 2020. Partitioning climate projection uncertainty with multiple large ensembles and CMIP5/6. *Earth System Dynamics*. 11(2): pp.491–508.

Padrón RS, Gudmundsson L, Decharme B, et al. 2020. Observed changes in dry-season water availability attributed to human-induced climate change. *Nature Geoscience*. 13(7): pp.477–481.

Qian C and Zhang X. 2015. Human Influences on Changes in the Temperature Seasonality in Mid- to High-Latitude Land Areas. *Journal of Climate*. 28(15): pp.5908–5921.

Vionnet V, Mortimer C, Brady M, et al. 2021. Canadian historical Snow Water Equivalent dataset (CanSWE, 1928–2020). *Earth System Science Data*. 13(9): pp.4603–4619.

Williams AP, Cook BI and Smerdon JE. 2022. Rapid intensification of the emerging southwestern North American megadrought in 2020–2021. *Nature Climate Change*. 12(3): pp.232–234.

Williams AP, Cook ER, Smerdon JE, et al. 2020. Large contribution from anthropogenic warming to an emerging North American megadrought. *Science*. 368(6488): pp.314–318.

Yao F, Livneh B, Rajagopalan B, et al. 2023. Satellites reveal widespread decline in global lake water storage. *Science*. 380(6646): pp.743–749.

Reviewer Reports on the First Revision:

Referees' comments:

Referee #1 (Remarks to the Author):

The manuscript is now substantially improved and all concerns that I have raised are well considered. In particular, the in situ observational data set of the revised paper includes 2,961 stations whereas the geographically limited data set in the original submission only incorporated some 550 stations. This makes the validity of the analysis much more convincing than that in the original submission. The temporal coverage of the new data set is also better, which tackles the problem of extrapolation, especially related to the long-term trends.

The methodology is also revised, apparently providing better overall confidence. One can argue, e.g., about the limitations and problems of ensemble analysis. Nevertheless, I think that this issue is better addressed and analyzed in the paper. Especially in complex terrain the use of ensemble approach is probably desirable, and it is useful to consider the consistency of different gridded dataset in the analysis. I am quite convinced that the conclusions of the paper regarding the human-forced changes in the snow cover are now justified new findings, and hence the paper deserves to be published.

Referee #2 (Remarks to the Author):

I would like to commend the authors on the revisions made to the analysis of this paper. All of my previous comments have been satisfactorily addressed with the new approach. I believe that the manuscript is now acceptable for publication from my perspective.

Referee #3 (Remarks to the Author):

I thank the authors for their thorough response to my previous points. They have made substantial improvements in the methods as well as datasets, and I think that the revised manuscript provides a robust attribution of snowpack changes to human influences. First, they made an observation-model comparison of hemispheric patterns of SWE trends and showed a clear emergence of anthropogenic signal in the observed SWE changes from the noise range of internal variability. Second, for the river-basin scale attribution, they reconstructed SWE by training a Random Forest model using all spatiotemporal data of temperature and precipitation rather than using local fitting. This technique greatly improved the reconstruction skill of their empirical model, which helps to increase the attribution confidence at river-basin scales. Furthermore, they performed a "perfect model" evaluation to check the validity of their approach to counterfactual reconstructions. The test results show that forced responses of SWE estimated based on temperature and precipitation changes can capture the "true" values reasonably. The authors also supported the robustness of their findings by

adding a few in situ observations and checking the stability of their empirical model. I have a couple of minor suggestions that the authors can consider for further clarification.

1. It would be useful to provide some references that support (1) the use of Random Forest algorithm compared to other methods if any and (2) the advantage of using all data rather than local data. I think that the authors can discuss the latter by showing how reconstruction skills change when using local data only in the same Random Forest model.

2. Using the “perfect model” analysis, the authors compared the forced trends in SWE estimated from the reconstruction model with the corresponding “true” forced trends (Fig. S16). However, they did not show whether their reconstruction approach works well in the future projections. I think a similar “perfect model” test can be done for future SWE estimates using SSP3-7.0 simulations.

Author Rebuttals to First Revision:

We thank the Reviewers and the Editor for their thoughtful engagement with this work. This Response to the Reviewers file provides a complete documentation of the changes that have been made in response to each individual Reviewer comment.

Reviewer comments are shown in plain text. Author responses are shown in bold text. Quotations from the revised manuscript are shown in bold italics. Line numbers in the author responses refer to locations in the revised manuscript.

Referee #1 (Remarks to the Author):

The manuscript is now substantially improved and all concerns that I have raised are well considered. In particular, the in situ observational data set of the revised paper includes 2,961 stations whereas the geographically limited data set in the original submission only incorporated some 550 stations. This makes the validity of the analysis much more convincing than that in the original submission. The temporal coverage of the new data set is also better, which tackles the problem of extrapolation, especially related to the long-term trends.

The methodology is also revised, apparently providing better overall confidence. One can argue, e.g., about the limitations and problems of ensemble analysis. Nevertheless, I think that this issue is better addressed and analyzed in the paper. Especially in complex terrain the use of ensemble approach is probably desirable, and it is useful to consider the consistency of different gridded dataset in the analysis. I am quite convinced that the conclusions of the paper regarding the human-forced changes in the snow cover are now justified new findings, and hence the paper deserves to be published.

We thank the Reviewer again for the suggestions to incorporate more long-term *in situ* data into the analysis, which has strengthened the methods and results.

Referee #2 (Remarks to the Author):

I would like to commend the authors on the revisions made to the analysis of this paper. All of my previous comments have been satisfactorily addressed with the new approach. I believe that the manuscript is now acceptable for publication from my perspective.

We thank the Reviewer for their thoughtful feedback that has improved the clarity and robustness of our methods, findings, and their interpretation.

Referee #3 (Remarks to the Author):

I thank the authors for their thorough response to my previous points. They have made substantial improvements in the methods as well as datasets, and I think that the revised manuscript provides a robust attribution of snowpack changes to human influences. First, they made an observation-model comparison of hemispheric patterns of SWE trends and showed a clear emergence of anthropogenic signal in the observed SWE changes from the noise range of internal variability. Second, for the river-basin scale attribution, they reconstructed SWE by training a Random Forest model using all spatiotemporal data of temperature and precipitation rather than using local fitting. This technique greatly improved the reconstruction skill of their empirical model, which helps to increase the attribution confidence at river-basin scales. Furthermore, they performed a “perfect model” evaluation to check the validity of their approach to counterfactual reconstructions. The test results show that forced responses of SWE estimated based on temperature and precipitation changes can capture the “true” values reasonably. The authors also supported the robustness of their findings by adding a few in situ observations and checking the stability of their empirical model. I have a couple of minor suggestions that the authors can consider for further clarification.

We thank the Reviewer for their thoughtful suggestions, which alongside the other Reviewer comments have increased the robustness and clarity of our original conclusions.

1. It would be useful to provide some references that support (1) the use of Random Forest algorithm compared to other methods if any and (2) the advantage of using all data rather than local data. I think that the authors can discuss the latter by showing how reconstruction skills change when using local data only in the same Random Forest model.

We agree. We now add additional references supporting the use of a Random Forest model in the Methods section:

“As another means of attributing historical SWE change, and to better understand its patterns and drivers at scales more commensurate with the impacts of snow loss, we generate a large observations-based ensemble of historical March SWE with and without the effects of anthropogenic forcing. We do so by using the common Random Forest machine learning algorithm, which fits randomized regression trees on bootstrapped samples of the data and averages their predictions together. The decision tree framework is particularly well-suited to pick up nonlinear interactions, such as that between temperature and precipitation in the context of snow, as well as correlated predictors. The Random Forest algorithm has been applied to reconstruct a wide variety of biogeophysical variables that are shaped by temperature, precipitation, and their interaction, including historical runoff (Ghiggi et al., 2019), crop yields

(Vogel et al., 2019), and climate-induced species range shifts (Lawler et al., 2006). In each instance, the Random Forest model was found to significantly outperform both other machine learning algorithms and more traditional approaches such as linear regression. Additionally, for this particular application of reconstructing historical snowpack, the model imposes no prior assumptions about temperature thresholds for rain-snow partitioning or snowmelt, which can vary substantially in space and are themselves a contributor to uncertainty in modeled estimates of SWE (Kim et al., 2021; Jennings et al., 2018).” (ll. 570-585)

We agree with the Reviewer’s second point about the relative value of using all data versus local data. We have retrained Random Forest models for each basin using only local data and compared the local model skill to that from the full panel. We now report the sensitivity analysis in the Methods section. We find that in most basins there are no significant differences in skill between the local and global models (based on the same significance criteria in the main). The local model does exhibit substantial degradations in skill in many of the highly-populated basins of the Western U.S., Western Europe, and High Mountain Asia, however. Combined with the fact that a panel model offers more data support for out-of-sample prediction, we feel the case for the full panel we use in the manuscript is strong. We have updated the Methods text to better communicate this important point:

“We fit the model using the full spatiotemporal panel of 0.5°x0.5° gridded data (i.e., all grid cell-years from 1981 to 2020), then aggregate the predicted gridded values to the river basin scale. We find that training a single model on the full panel of data offers two main advantages over training multiple models on more local data (e.g., a model for each river basin). First is that the out-of-sample prediction skill of the full panel model is significantly higher in many highly-populated mid-latitude basins of the Western U.S., Western Europe, and High Mountain Asia; local models are more skillful in fewer than 20 percent of basins, concentrated in sparsely-populated high-latitude basins where the skill of the full panel model is already high (Extended Data Fig. 3). Second, training a single model on data from the entire hemisphere provides greater statistical stability of projections made with large perturbations to the input variables, such as adding an end-of-century climate change signal (Extended Data Fig. 8), which could exceed the support of local historical observations as records fall at an increasing rate (Coumou, Robinson, and Rahmstorf, 2013; Rahmstorf and Coumou, 2011).” (ll. 591-603)

2. Using the “perfect model” analysis, the authors compared the forced trends in SWE estimated from the reconstruction model with the corresponding “true” forced trends (Fig. S16). However, they did not show whether their reconstruction approach works well in the future projections. I think a similar “perfect model” test can be done for future SWE estimates using SSP3-7.0 simulations.

We thank the Reviewer for this suggestion and have plans to pursue a thorough analysis of the snowpack projections given our reconstructions in future work.

Response References

Coumou D, Robinson A and Rahmstorf S. 2013. Global increase in record-breaking monthly-mean temperatures. *Climatic Change*. 118(3): pp.771–782.

Ghiggi G, Humphrey V, Seneviratne SI, et al. 2019. GRUN: an observation-based global gridded runoff dataset from 1902 to 2014. *Earth System Science Data*. 11(4): pp.1655–1674.

Jennings KS, Winchell TS, Livneh B, et al. 2018. Spatial variation of the rain–snow temperature threshold across the Northern Hemisphere. *Nature Communications*. 9(1): pp.1148.

Kim RS, Kumar S, Vuyovich C, et al. 2021. Snow Ensemble Uncertainty Project (SEUP): quantification of snow water equivalent uncertainty across North America via ensemble land surface modeling. *The Cryosphere*. 15(2): pp.771–791.

Lawler JJ, White D, Neilson RP, et al. 2006. Predicting climate-induced range shifts: model differences and model reliability. *Global Change Biology*. 12(8): pp.1568–1584.

Rahmstorf S and Coumou D. 2011. Increase of extreme events in a warming world. *Proceedings of the National Academy of Sciences*. 108(44): pp.17905–17909.

Vogel E, Donat MG, Alexander LV, et al. 2019. The effects of climate extremes on global agricultural yields. *Environmental Research Letters*. 14(5): pp.054010.